# Violet/ultraviolet light-induced depassivation in halide perovskite solar cells

Dejian Yu[1,6], Fei Cao[1,6], Xiaosong Qiu[1,6], Ying Chen[1,6], Zhipeng Zhang[1], Gang Wang[1], Yulin Mao[1], Junwen Zhong [2] ✉, Chenliang Su [3] ✉, Wei Huang [4], Tae-Woo Lee [5] ✉ & Guichuan Xing [1] ✉

Ammonium-terminated ligands, used directly or in the form of two-dimensional perovskites, are leading defect-passivating agents for halide perovskites and have significantly contributed to achieving the highest efficiencies across diverse perovskite solar cells. However, even these state-of-the-art perovskite solar cells suffer from rapid degradation during operation, raising concerns over the durability of the passivation. In this work, we unveil a mechanism of violet/ultraviolet light-induced depassivation that universally affects ammonium ligands. Exposure to violet/ultraviolet light triggers a charge-carrier transition from ammonium-terminated ligands to the halide perovskite framework, leading to the deprotonation of the ammonium group. This deprotonation disrupts the ammonium-perovskite interaction, resulting in depassivated perovskites susceptible to photodegradation, as validated in cells with 26.44% efficiency. This updated understanding surpasses existing ligand failure models limited to thermal intolerance and iodide oxidation, highlighting an essential perspective for enhancing the long-term efficacy of passivating agents.

Halide perovskite (HP) solar cells have garnered significant research interest over the past decade, achieving a remarkable power conversion efficiency (PCE) of 27.0% in single-junction, small-area configurations[1]. One key factor in this improvement is the effective passivation of defects. Although defects in HPs are relatively benign compared to those in traditional semiconductors, they still considerably limit the photovoltaic performance[2,3]. Significantly, defects also drive degradation by facilitating ion migration[4], environmental erosion[5], and phase transition[6]. To combat these issues, passivating agents that interact with and neutralize defects are essential.

Ammonium-terminated ligands (R-NH$_3^+$, R represents alkyl or aryl groups) have been the leading defect-passivating agents. They can serve directly as passivating agents themselves, acting as Lewis acids to neutralize negatively charged defects such as common uncoordinated halogen anions and Pb-I anti-sites, through hydrogen bonding and Coulomb interactions[7,8]. Alternatively, they are widely used to generate two-dimensional (2D) HP derivatives, which offer passivation and affect carrier extraction at the 3D/2D HP interface[9–11]. Merits like flexible molecular structure[12], strong chemical affinity for HPs[13], broad adaptability across HPs with diverse compositions, sizes, dimensionalities, and bandgaps[14–17], as well as excellent reproducibility in laboratories worldwide, set ammonium ligands apart. They have contributed to realizing the highest PCEs in various solar cell types, including (lead HP) p-i-n cells[18], (lead HP) n-i-p cells[19], tin-lead HP

---

[1]Institute of Applied Physics and Materials Engineering, University of Macau, Macau, China. [2]Department of Electromechanical Engineering, Faculty of Science and Technology, University of Macau, Macau, China. [3]International Collaborative Laboratory of 2D Materials for Optoelectronics Science and Technology of Ministry of Education, Institute of Microscale Optoelectronics, Shenzhen University, Shenzhen, China. [4]MIIT Key Laboratory of Flexible Electronics (KLoFE), Institute of Flexible Electronics (IFE), Northwestern Polytechnical University (NPU), Xi'an, China. [5]Department of Materials Science and Engineering, Institute of Engineering Research, Research Institute of Advanced Materials, Soft Foundry, Interdisciplinary Program in Bioengineering, Seoul National University, Seoul, Republic of Korea. [6]These authors contributed equally: Dejian Yu, Fei Cao, Xiaosong Qiu, and Ying Chen. ✉e-mail: junwenzhong@um.edu.mo; chmsuc@szu.edu.cn; twlees@snu.ac.kr; gcxing@um.edu.mo

cells[20], pure-tin HP cells[21], all-HP tandem cells[22], HP/silicon tandem cells[17], HP/organic tandem cells[23], HP/CuInSe$_2$ tandem cells[24], large-area HP solar modules (715.1 cm$^2$)[25], indoor HP photovoltaics[26], and air-fabricated HP cells[27].

However, contradictory to the excellent defect passivation by ammonium ligands, rapid degradation during operation remains a critical challenge for these state-of-the-art HP cells, which raises concerns about the durability of ammonium ligands. Increasing attention is paid to studying the depassivation of ammonium ligands. Meng et al. showed that deprotonation of organic cations can occur in the precursor stage with the oxidation of iodide ions[28]. Wang et al. reported that ammonium ligands with a low acid dissociation constant can be thermal-intolerant and might fail passivation under high-temperature light soaking[29]. The poor thermal/light tolerance of typical ammonium ligands was also confirmed by Pei et al.[30]. However, the depassivation mechanisms of ammonium ligands are still insufficiently understood and warrant further exploration.

In this work, we reveal a violet/ultraviolet (UV) light-driven depassivation mechanism that universally affects ammonium-terminated ligands. Contrary to the common belief that these ligands play a negligible role in photogenerated carrier dynamics, we find that violet/UV photons induce an electron transition from the ammonium group to the HP framework, which subsequently causes the deprotonation of the ammonium group. This deprotonation disrupts the ammonium-HP interaction and leads to depassivated HP susceptible to photodegradation, as validated in cells with 26.44% PCE. These results highlight the vulnerability of the prevailing passivation strategy with ammonium ligands toward sunlight, providing crucial insights into the development of more robust solutions.

## Results

### Ligand-HP interfacial carrier dynamics

To effectively investigate the proposed mechanism, a sample with a high density of well-ordered ammonium-HP bonds is essential. The ammonium-passivated 3D HP system is not an ideal platform due to the sparse and disordered distribution of the ligands on the 3D HP surface[31]. Instead, we utilized Ruddlesden-Popper 2D HPs with the formula L$_2$PbI$_4$, where L represents ammonium-terminated ligands. As exemplified by BA$_2$PbI$_4$ (BA denotes butylammonium, Fig. 1A), the -NH$_3^+$ head of the BA ligands occupies positions typically held by cesium, methylammonium, or formamidinium in 3D HPs, connecting to the HP framework via the (-NH$_3^+$)·I$^-$ bonding in the same manner as in ammonium-passivated 3D HPs[32]. The BA ligands assemble orderly on the two sides of the HP framework, creating atomically dense and organized ammonium-HP connections to make L$_2$PbI$_4$ an ideal sample for mechanistic exploration. Moreover, the well-defined ammonium-HP interface enables a clear electronic structure that facilitates characterization.

Notably, in L$_2$PbI$_4$, both the conduction band minimum (CBM) and valence band maximum (VBM) arise from the inorganic HP sublattice[33]. In contrast, ammonium-terminated ligands like BA act as electronic barriers, confining charge carriers within the HP sublattice and forming quantum wells. Therefore, these ligands have been considered to have negligible participation in photogenerated carrier dynamics[34]. However, our experimental findings deviate from this interpretation. To characterize the carrier dynamics of a BA$_2$PbI$_4$ film, broadband ultrafast transient absorption (TA) spectroscopy was employed, using a pump energy of 3.82 eV (325 nm). The well-defined structure of BA$_2$PbI$_4$ is confirmed by the X-ray diffraction (XRD) pattern (Fig. S1), the steady-state absorption spectrum (Fig. S2), and the photoluminescence (PL) spectrum (Fig. S2). As shown in Fig. 1B, three distinct photobleaching (PB) signals can be identified. The PB signal in the range of 2.43–2.57 eV corresponds to the inherent excitonic absorption of BA$_2$PbI$_4$, resulting from transitions between the VBM and the exciton state (E$_x$)[35]. However, the additional two PB signals

observed in the near-UV range of 2.90–3.10 eV and 3.20–3.40 eV are rarely reported, and their origin remains unclear, warranting further investigation.

We first rule out the possibility that the near-UV PB signals arise from raw reagents, i.e., PbI$_2$ or BAI, as their absorption peak positions do not match the observed PB signals (Fig. S3)[36]. Furthermore, we conducted TA characterization on a series of 2D HP films made with various ammonium-terminated ligands, including hexylammonium (HA), octylammonium (OA), dodecylammonium (DA), phenylethyl-lammonium (PEA), and phenylpropylammonium (PPA). These ligands feature aliphatic and aromatic side groups with varying chain lengths. The results reveal that the near-UV PB signals are not unique to BA$_2$PbI$_4$ but are a universal feature across these 2D HP systems (Figs. S4–S9). The XRD patterns (Fig. S4) suggest that some of these 2D HP films may contain minimal unidentified impurities, but these impurities are inconsistent across all films. Other 2D HPs such as PPA$_2$PbI$_4$ and OA$_2$PbI$_4$ are highly phase-pure with no discernible impurities. These observations indicate that the near-UV PB signals do not arise from specific impurities or 2D HP/impurity heterojunctions. Therefore, we infer that the near-UV PB signals originate directly from the 2D HPs.

The lower-energy near-UV PB signal's change in absorbance (ΔA) is comparable to that of the excitonic PB signal, indicating significant transition strength, while the higher-energy PB signal's ΔA is weaker. Notably, we notice a subtle but discernible correlation between the lower-energy near-UV PB signal and the excitonic PB signal. The initial decay of the near-UV PB signal coincides with a slight rise of the excitonic PB signal (Fig. 1C). This suggests a charge transfer process from certain higher energy states to the VBM or the E$_x$. Therefore, the near-UV PB signal is unlikely to stem from excited-state absorption or the Auger process, as both mechanisms promote transitions to high-energy levels by depleting excited-state carriers, which would result in a reversed rise-decay relationship. Furthermore, even upon quasi-resonance excitation for the excitonic absorption (using pumping light at 2.48 eV, 500 nm), the near-UV PB signals featuring higher energies are detected as anti-Stokes PB signals across all the 2D HPs (Fig. S10). Thus, the near-UV PB signals could arise from carrier transitions involving the VBM or E$_x$. The observation also excludes the scenario that the near-UV PB signals are from hot carriers. Otherwise, it should be unobservable upon the lower-energy excitation.

To better understand the origin of the near-UV PB signal, we quantitatively depict the electronic structure of BA$_2$PbI$_4$ with a high-light of key energy levels, including the CBM, E$_x$, VBM, lowest unoccupied molecular orbital (LUMO) of the ligand layer, and highest occupied molecular orbital (HOMO) of the ligand layer, in Fig. 1D. The VBM is measured with ultraviolet photoelectron spectroscopy (UPS) characterization (Fig. S11). The obtained VBM (−5.82 eV) and the accordingly inferred CBM based on a bandgap of 2.7 eV agree with the previous report by Silver et al[37]. E$_x$ is determined per the excitonic absorption energy (Fig. S2, 2.41 eV). The HOMO and LUMO also refer to the report by Silver et al. obtained with density functional theory (DFT) calculations[37]. Based on the energy diagram, the energies of the two near-PB signals (2.9–3.1 eV for the low-energy one and 3.2–3.4 eV for the higher-energy one) closely match those of the HOMO-to-E$_x$ (3.1 eV) and HOMO-to-CBM (3.4 eV) carrier transitions, respectively. The minor deviations are reasonable considering that the DFT calculation adopted for the HOMO in the previous report usually overvalues the result. A supportive piece of evidence might be the emergence of the onset of a new band in the Fermi region of the UPS spectrum at approximately 0.43 eV below the VBM (Fig. S11), which can tentatively be attributed to the HOMO. As such, the true HOMO might be within the range of −6.25 eV (−5.82 eV minus 0.43 eV) to −6.50 eV versus the vacuum level, so the HOMO-E$_x$ gap and the HOMO-CBM gap could be in the range of 2.85–3.10 eV and 3.15–3.40 eV, justifying the above deviations. Notably, the HOMO-to-E$_x$ and HOMO-to-CBM transitions are also consistent with the above observation of anti-Stokes PB signals.

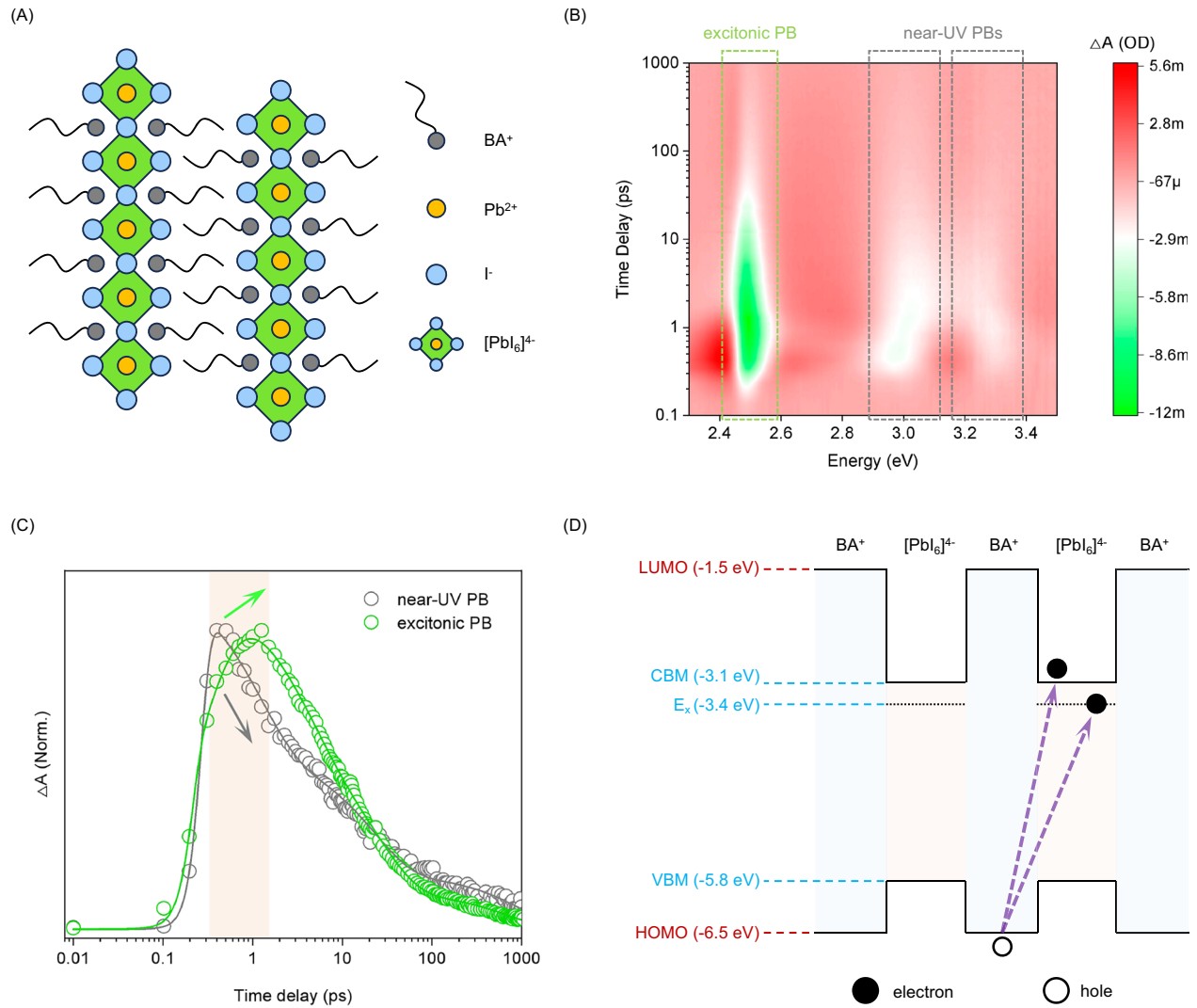

**Fig. 1 | Ultrafast spectroscopic evidence for the ligand-to-HP interfacial electron transition. A** Crystalline structure diagram of $BA_2PbI_4$, which is used as the platform for mechanistic exploration. **B** Pseudocolor ultrafast TA spectrum of the $BA_2PbI_4$ film under 3.82 eV (325 nm, 1 kHz, 100 fs, ~510 μW/cm²) laser excitation, showing two near-UV PB signals in addition to the excitonic PB signal. **C** Corresponding decay kinetics of the two PB signals, monitoring the excitonic PB signal at 2.49 eV and the near-UV PB signal at 3.0 eV, respectively. **D** Schematic illustration of the electronic structure of $BA_2PbI_4$ with a highlight of key energy levels relative to the vacuum level. The LUMO and HOMO levels were obtained by DFT simulation in Silver's work[37]. Other energy levels were determined from experiments reported in the literature and in this work.

The HOMO-to-$E_x$/CBM transitions will generate a dipole at the ligand-HP interface due to the separation of electrons and holes. To investigate the formation of this interfacial dipole, we performed steady-state optoelectrical characterizations based on the following perspective: violet/UV photons have a limited penetration depth in $BA_2PbI_4$ films, approximately 113 nm for 400 nm photons[38]. Therefore, in a thick film where violet/UV photons cannot penetrate the entire thickness, photoexcitation occurs near the illuminated front surface but not at the unilluminated back surface. If photoexcitation induces the HOMO-to-$E_x$/CBM transition, dipoles will exist at the ligand-HP interface in the illuminated front, while the non-illuminated back will lack these dipoles. This creates a global potential difference (ΔV) across the thick film, as illustrated in Fig. 2A. Conversely, in a thin film where violet/UV photons can penetrate its entire thickness, photoexcitation occurs throughout the film, so this ΔV should be absent.

We then employed a thick film of ~337 nm and a thin film of ~29.5 nm (Fig. S12) for comparison. The $BA_2PbI_4$ films were tested in a vertical ITO/$BA_2PbI_4$/Cu device configuration. The indium tin oxide (ITO) and copper (Cu) electrodes have similar work functions (ITO in

the range of 4.6–4.75 eV and Cu at 4.7 eV), rendering energetic symmetry across the device. The $BA_2PbI_4$ films exhibit an out-of-plane orientation (Fig. S1), along which direction the ligand layer and the HP sublattice alternate.

The current-voltage (I–V) measurements for the ITO/$BA_2PbI_4$/Cu devices demonstrated an ohmic contact feature (Fig. S13), regardless of the $BA_2PbI_4$ film thickness. The presence of ΔV can be confirmed by analyzing the symmetry of the overall I–V characteristic. Specifically, an ideally symmetrical energy landscape without ΔV should result in a symmetrical I-V characteristic. In this case, the same absolute bias ($V_{abs}$) would generate equal currents in the forward and reverse directions (denoted as $I_{for}$ and $I_{rev}$, respectively), leading to a zero-sum of $I_{for}$ and $I_{rev}$ ($I_{for} + I_{rev} = 0$). Conversely, when ΔV is present, the ($I_{for} + I_{rev}$) versus $V_{abs}$ curve would display a diode-like characteristic.

In the dark, the 337 nm-thick $BA_2PbI_4$ film exhibits a quasi-exponential ($R^2 = 0.956$) decrease in the ($I_{for} + I_{rev}$) as the applied bias $V_{abs}$ increases (Fig. 2B). The imperfect exponential dependence indicates the presence of weak built-in potentials at contacts, which can be attributed to the slight difference in built-in potential between the

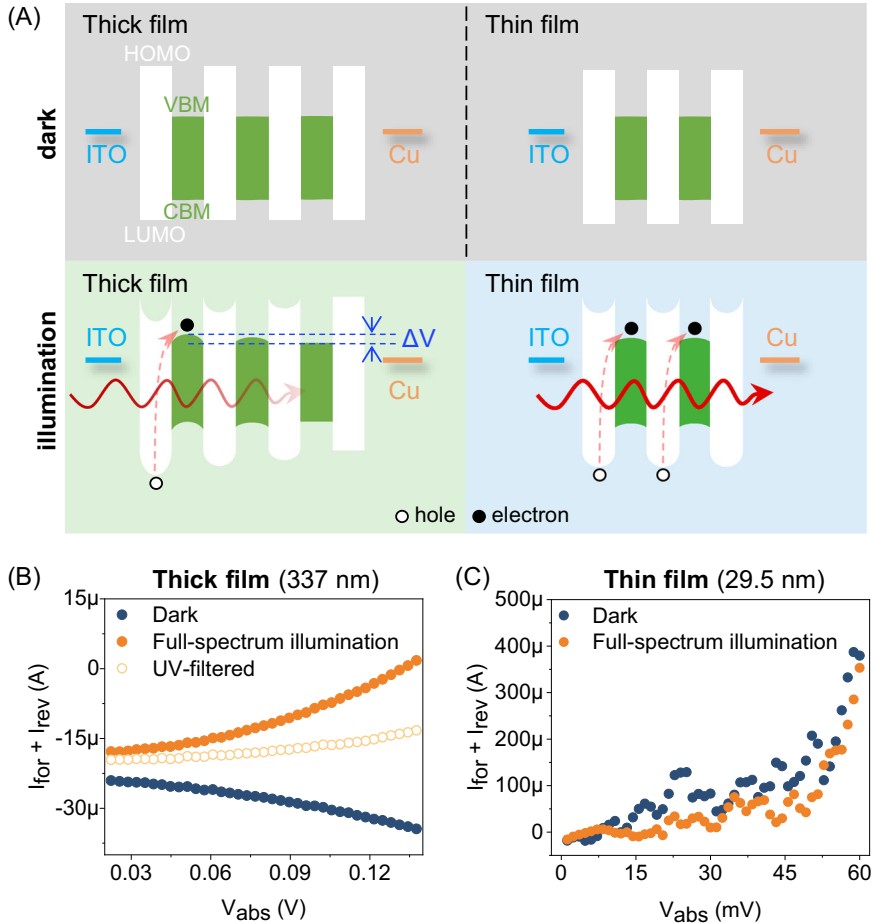

**Fig. 2 | Steady-state optoelectrical characterization of the ligand-HP interfacial electron transition. A** Schematic energy landscape for thick (left panels) and thin (right panels) $BA_2PbI_4$ films, shown in the dark (top panels) and under illumination (bottom panels). **B** I–V characteristics of the thick $BA_2PbI_4$ film in the dark, under full-spectrum one-sun illumination, and under one-sun illumination excluding violet/UV photons (energy > 2.95 eV). **C** I–V characteristics of the thin $BA_2PbI_4$ film in the dark and under full-spectrum one-sun illumination.

near-ITO side and the near-Cu side, as illustrated in Fig. S14. The built-in potential disparity is fitted to be 57.8 ± 7.2 meV, which aligns with the reasonable work function difference between the Cu and ITO electrodes. Intriguingly, this I–V dependence reverses upon simulated solar illumination. The $(I_{for} + I_{rev})$ instead increases exponentially with $V_{abs}$, and it follows a perfect exponential dependence ($R^2 = 0.993$), suggesting the emergence of an inverse $\Delta V$ upon illumination, which is consistent with the effect of the interfacial dipole. To further confirm that the photogenerated $\Delta V$ is induced by the violet/UV photons, a long-pass filter was used to pre-screen the simulated solar illumination, allowing only photons with wavelengths greater than 420 nm (energy <2.95 eV) to pass. Indeed, under this filtered illumination, the diode-like characteristic is significantly mitigated, indicating the absence of the photogenerated $\Delta V$. These observations confirm the presence of a global potential difference $\Delta V$ induced by violet/UV photons in the thick $BA_2PbI_4$ film. Notably, under the filtered illumination, the $(I_{for} + I_{rev})$ versus $V_{abs}$ characteristic does not revert to that in the dark. This is because the photoinjected carrier density is sufficient to eliminate the built-in potentials at both the near-Cu side and the near-ITO side of the film, as illustrated in Fig. S15.

In contrast to the thick film, the photogenerated $\Delta V$ is not observed in the 29.5 nm $BA_2PbI_4$ thin film. As shown in Fig. 2C, the $(I_{for} + I_{rev})$ versus $V_{abs}$ characteristics are consistent for being in the dark and under illumination. This contrast between the thick and thin $BA_2PbI_4$ films confirms the violet/UV light-induced HOMO-to-$E_x$/CBM transition that generates the interfacial dipole. Importantly, all the

above I-V measurements were double-tested to ensure data reliability (Figs. S16, S17). Furthermore, similar optoelectrical characterizations were conducted on a series of 2D HPs, which confirmed the universality of the observed results. These additional measurements on the 2D HP systems are presented in Figs. S18–S22.

## Violet/UV light-induced deprotonation of the ammonium group
The HOMO level in these 2D HPs is contributed by the C–H σ-bonds, C–C σ-bonds, and N–H σ-bonds[37,39]. The direct attachment of the ammonium group to the HP sublattice facilitates efficient orbital overlap[40], making the ammonium group the primary contributor to the HOMO-to-$E_x$/CBM electron transition process. In contrast, the aryl/alkyl group of the ligands extends outward and is not connected to the HP sublattice, unlikely to participate in the transition. Indeed, for 2D HPs by ammonium ligands with different aryl/alkyl groups, the energies of the near-UV PB signals are consistent (Figs. S5–S9), confirming the negligible impact of the aryl/alkyl groups. In the following, we term the HOMO-to-$E_x$/CBM electron transition the ammonium-to-HP electron transition to facilitate the discussion.

The ammonium group ($-NH_3^+$) is formed by protonating a neutral amine ($R-NH_2$) with a proton ($H^+$). The protonation occurs when the nitrogen atom in the amine group shares a lone pair of electrons with $H^+$, which results in three chemically equivalent N–H bonds within the ammonium group, with the positive charge delocalized across the entire structure. Thus, the ammonium-to-HP electron transition under violet/UV light could deprotonate the

ammonium group, disrupting its interaction with the HP sublattice. This mechanism is consistent with the well-documented easy degradation of 2D HPs upon UV exposure[41,42].

To confirm this deprotonation of the ammonium ligands due to the ammonium-to-HP electron transition, we designed a photoreaction system where the deprotonation of ammonium results in measurable $H_2$ production. Specifically, 2D HPs in the form of nanoplates are dispersed in hydroiodic acid (termed HI hereafter) to trigger the following reactions upon violet/UV illumination:

$$I^- + h^+ \rightarrow 1/2 I_2 \qquad (1)$$

$$1/2 R - NH_3^+ + 1/2 H^+ + e^- \rightarrow 1/2 R - NH_2 + 1/2 H_2 \uparrow \qquad (2)$$

The $H^+$ and $I^-$ ions are from the HI solution, and the proton in the $(-NH_3^+)$ group participates in the photoreactions. These photoreactions are framed by the DFT simulation results (Fig. S23, Supplementary Note 1). A crucial link between the photoreactions and the deprotonation dynamics of the ammonium group is that the nitrogen atom serves as the active site for both photoreactions (Fig. 3A), as the adsorption of $H^+$ and $I^-$ is most energetically favorable at the nitrogen location.

The two photoreactions work together in a cyclic mechanism that allows for a continuous production of $H_2$. Reaction (1) is endergonic ($\Delta G > 0$, $\Delta G$ refers to the Gibbs free energy change), while reaction (2) occurs spontaneously ($\Delta G < 0$). This means energy input from violet/UV photons is necessary to drive the cyclic process. The overall cycle of the violet/UV light-driven photoreaction can be summarized as follows (Fig. 3B):

1. Violet/UV illumination excites the ammonium-to-HP electron transition;
2. Then, the photogeneration of holes in the ammonium groups drives the endergonic oxidation of $I^-$ ions (as $I_3^-$) in the solution to $I_2$.
3. Subsequently, $H^+$ ions from the surrounding solution react with the ammonium groups to produce $H_2$.
4. Consequently, the ammonium groups are deprotonated to $-NH_2$ moieties;
5. Finally, the $-NH_2$ moieties are re-protonated by $H^+$ ions in the solution, resetting the cycle to the initial state.

Accordingly, the production of $H_2$ is a direct result of the dynamic deprotonation of the ammonium groups. The vessel used for the photoreactions was evacuated to minimize interference from other potential oxidants and reductants. The temperature of the vessel was fixed at 10 degrees Celsius during the test to prevent interference from heating. The 2D HP is stable throughout the test as discussed in the Supporting Information file (Fig. S24, Supplementary Note 2).

$H_2$ production is only detectable under illumination, confirming that photoexcitation is essential for initiating hydrogen generation. Using an Xe lamp as the light source (250 mW cm$^{-2}$), we measured $H_2$ production rates under two different illumination conditions, as shown in Fig. 3C: (1) filtered illumination, which excludes violet/UV photons with wavelengths below 420 nm (approximately 5% of the total illumination power); and (2) full-spectrum, unfiltered illumination. The experimental results reveal a significant enhancement in the hydrogen evolution activity upon the full-spectrum, unfiltered illumination. Specifically, the approximately 5% of violet/UV power not filtered out leads to a dramatic 186% increase in the hydrogen evolution rate, rising from 1.69 mmol g$^{-1}$ h$^{-1}$ to 4.83 mmol g$^{-1}$ h$^{-1}$. This finding highlights the vital role of violet/UV photons in driving the photoreaction cycle, confirming the violet/UV light-induced deprotonation of the ammonium groups in 2D HPs. The minor degree of $H_2$ production upon visible illumination (photon wavelength >420 nm) could

arise from the photocatalytic $H_2$ generation by the inorganic HP moiety. Specifically, the direct transfer of photogenerated holes and electrons from the inorganic HP framework to the $I^-/H^+$ ions in the solution may contribute to this residual $H_2$ evolution (Fig. S25, Supplementary Note 3)[43].

In contrast to in the HI solution, re-protonation is unlikely to occur in solid-state solar cells due to the absence of free $H^+$ ions. Consequently, deprotonation of the ammonium group would disrupt the ammonium-HP interaction, leading to the degradation of 2D HPs. A detailed analysis of the violet/UV light-induced degradation of the 2D HP $BA_2PbI_4$ is provided in Fig. S26, Supplementary Note 4. This degradation pathway underscores a potential failure mechanism for the commonly employed strategy of utilizing 2D HPs as passivating agents in HP solar cells.

## Violet/UV light-induced depassivation in HP solar cells

Since 2D and 3D HPs exhibit the same chemical interactions between ammonium-terminated ligands and the HP framework, the violet/UV light-activated ammonium-to-HP electron transition and the resulting disrupted ligand-HP interaction should also apply to ammonium-passivated 3D HPs. To explore this concept in the 3D HP system, we first conducted TA spectrum measurements on a multiphasic quasi-2D HP film. This film follows the general formula $BA_2(CH_3NH_3)_4Pb_5I_{16}$ and contains both 2D and 3D HP phases with a gradient of dimensionality[44,45]. We used excitation light of 2.48 eV (500 nm) to photoexcite all the 2D and 3D HP phases. As shown in Fig. 4A, PB signals were observed for $n = 3, 4, 5,$ and $> 5$ phases, consistent with the absorption spectrum (Fig. S27). Importantly, a broadened PB signal is detected in the violet and UV regions. The energy of this broadened PB signal is higher than the 2.48 eV excitation light, which aligns with the proposed ammonium-to-HP electron transition mechanism. The broadening of the PB signal can be attributed to a series of energy gaps between the organic HOMO and inorganic $E_x$/CBM in the multiphasic film. Consistently, the broadened PB signal is also observed in a series of quasi-2D HPs with various ammonium-terminated ligands, confirming the generality of this finding (Figs. S28–S31). These results demonstrate that the violet/UV light-activated ammonium-to-HP electron transition applies to quasi-2D HPs with increasing dimensionality and is reasonably extendable to 3D HPs.

To support this point, we compared the absorption spectra of a pristine 3D HP ($MAPbI_3$, MA denotes methylammonium) film and a surface-treated 3D HP film with BA in Fig. 4B. To avoid interference from 2D HPs, the dose of ammonium-terminated ligands (BAI in this case) for surface treatment was carefully controlled. The absence of 2D HPs is demonstrated in Fig. S32. The two absorption spectra exhibit perfect overlap within the tested wavelength range, except between 300 and 400 nm (corresponding to photon energies between 3.10 eV and 4.13 eV), where the absorption of the ligand-treated film is enhanced compared to the pristine film. Since $MAPbI_3$ is mono-cationic (containing only the $CH_3NH_3^+$ cation, no $Cs^+$ or $CH(NH_2)_2^+$) and mono-anionic (containing only the $I^-$ anion), the observed enhanced absorption is unlikely to arise from phase separation. Moreover, the enhanced absorption does not originate from BAI (as evidenced in Fig. S3). Instead, the energy range of the enhanced absorption coincides with the violet/UV light-induced ammonium-to-HP electron transition mechanism. Furthermore, consistent with the universality of the mechanism, the enhanced absorption in the 300 to 400 nm range is also observed in 3D HP films treated with other types of ammonium-terminated ligands, as shown in Fig. S33. These results confirm the applicability of the proposed mechanism to 3D HPs. Notably, resolving the ammonium-to-HP electron transition signal in surface-treated 3D HP films using TA characterization is challenging due to its overlap with the intrinsic high-energy absorption signals of 3D HPs, as well as difficulties in eliminating interference from 2D HPs (Fig. S34).

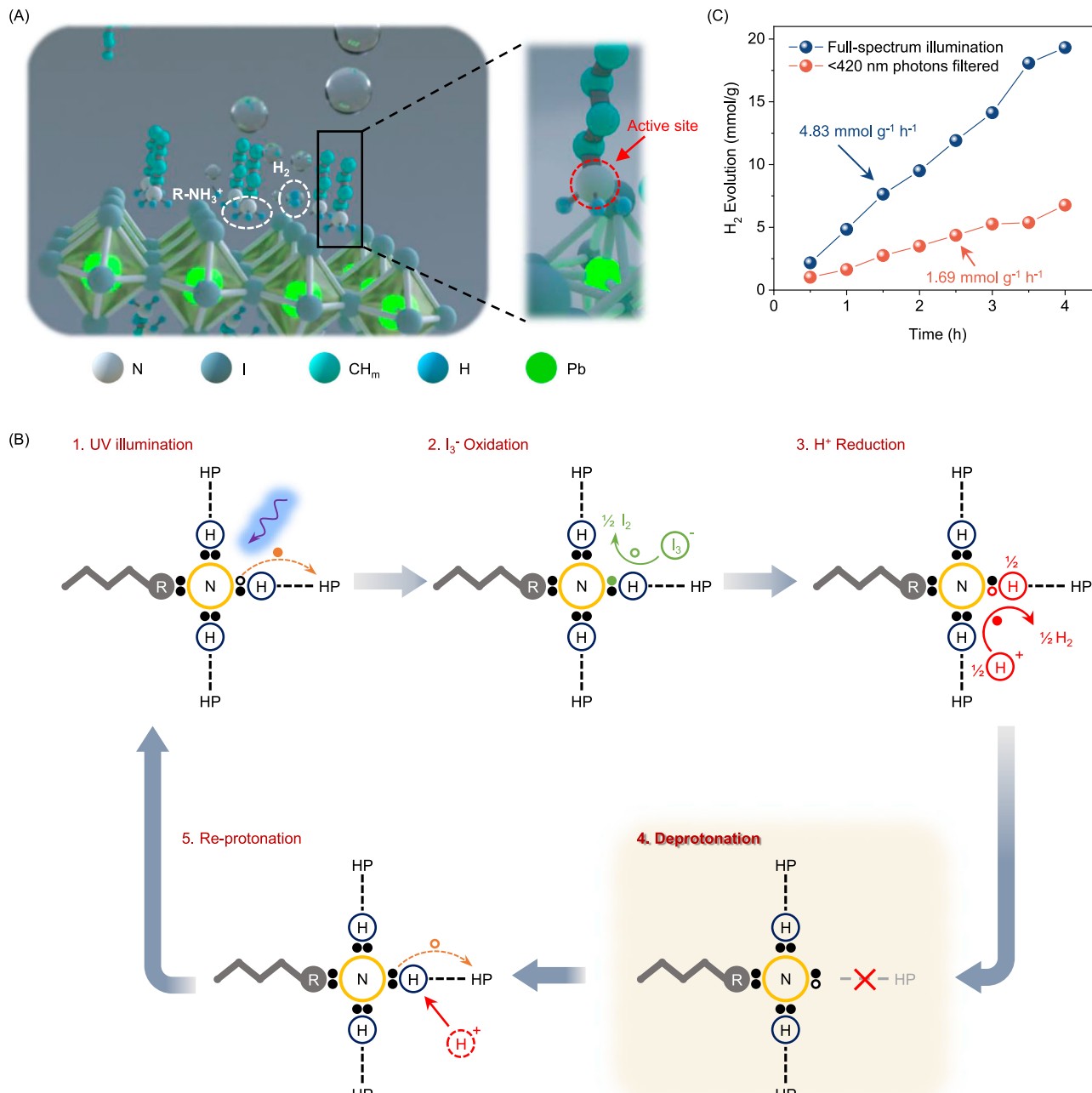

**Fig. 3 | Violet/UV light-induced deprotonation of the ammonium group.**
**A** Schematic diagram of the designed photoreaction, in which deprotonation of the ammonium ligands produces $H_2$. The nitrogen atom of the ammonium ligand serves as the active site for this process. **B** Diagram of the violet/UV light-triggered photoreaction cycle. Hollow circles denote holes, solid circles denote electrons, and colored holes/electrons denote the charge carriers participating in the corresponding kinetics. **C** Hydrogen generation activities of the 2D HP under full-spectrum illumination and violet/UV light-filtered illumination (< 420 nm photons filtered out), respectively.

Consequently, exposure to violet/UV photons causes the deprotonation of ammonium ligands, disrupting the ammonium-HP interaction. To verify this, we conducted Fourier-transform infrared (FTIR) spectroscopy on both UV-irradiated (365 nm @ 25 mW/cm² for 5 h) and non-irradiated ammonium-passivated MAPbI₃ samples. During irradiation, the 3D HP powder was gently stirred and dispersed in chlorobenzene with ligands. It is well established that HPs adopt an X-type ligand binding configuration, where Lewis acid and Lewis base moieties attach to the surface of HP in pairs[31,46]. Herein, we employed BA and oleic acid (1:1 in molar ratio) as the ligand pair. The 3D HP powder was collected every hour, followed by isolation for the FTIR characterization. As shown in Fig. 4C, the 2854 cm⁻¹ peak arises from the symmetric stretching vibration of -CH₂- groups. It originates from both the butylammonium and oleic acid segments in the ligand pair but is not detected in MAPbI₃. Notably, as the irradiation time increases, the -CH₂- peak wanes gradually, indicating the detachment of ligands from the HP surface. After 5 h of irradiation, this characteristic peak becomes barely discernible. In contrast, for the non-irradiated sample, the 2854 cm⁻¹ peak remains after the 5 h test. This contrast confirms that the 365 nm irradiation is the driving factor for ligand detachment, in agreement with our mechanism.

These results show that violet/UV photons in natural solar illumination will nullify the passivation effect of ammonium ligands in HP solar cells. The depassivated HP could be susceptible to defect

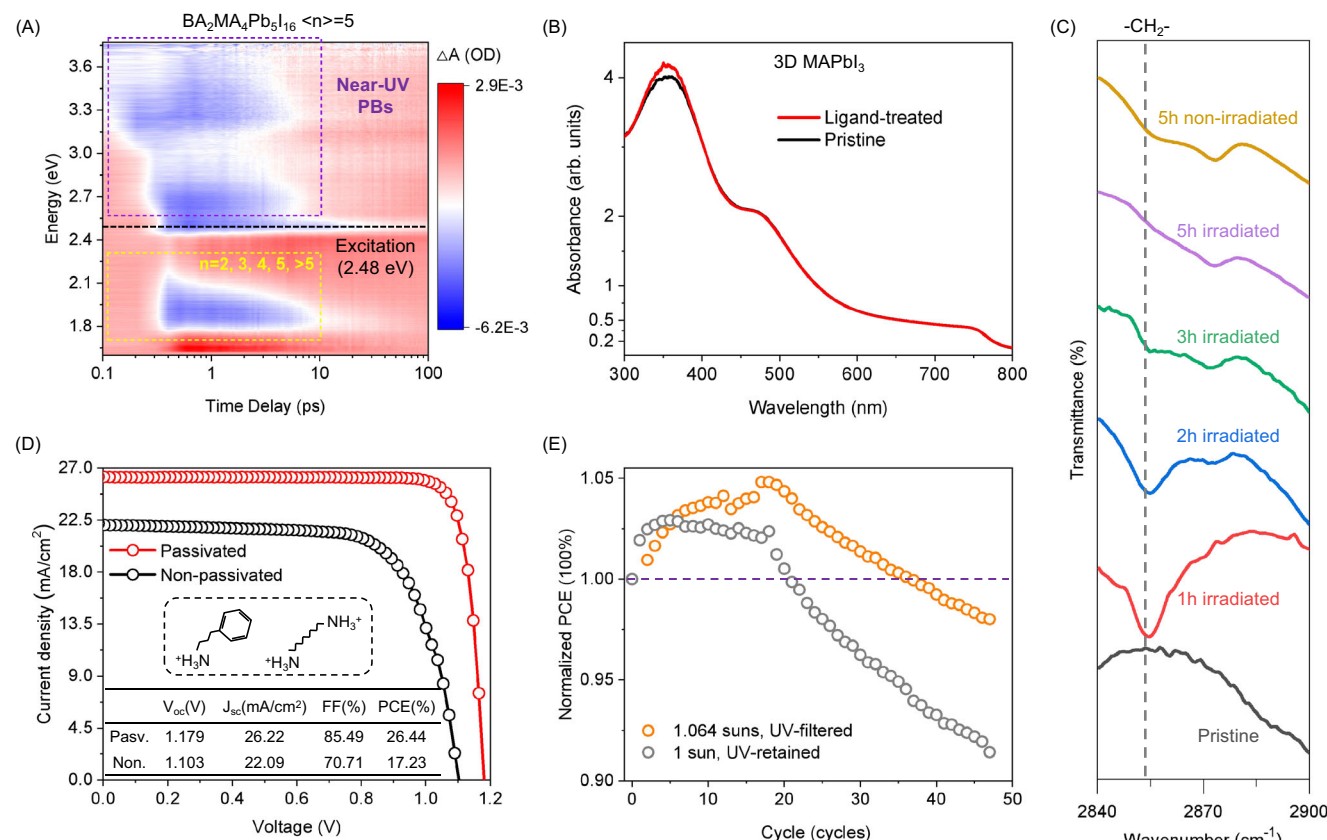

**Fig. 4 | Violet/UV light-induced depassivation in HP solar cells. A** Pseudocolor ultrafast TA spectrum of a multiphasic $BA_2MA_4Pb_5I_{16}$ film with a nominal <n> value of 5. The excitation energy was set at 2.48 eV (500 nm, 1 kHz, 100 fs). **B** Comparative absorption spectra of 3D HP films before and after surface treatment with ammonium ligands. **C** Comparative FTIR spectra of ammonium-passivated 3D HPs subjected to UV irradiation (365 nm @~25 mW/cm², for 5 h) and in the dark. **D** I–V characteristics of passivated and non-passivated solar cells, along with a summary

of their device performance. Insets: molecular structures of the ammonium ligands used for passivation. The left one is PPA, and the right one is octane-1,8-diammonium. **E** Comparative PCE evolutions of ammonium-passivated solar cells upon violet/UV light-contained and violet/UV light-filtered solar illuminations. The solar cells were subjected to a light-dark alternation test with 7 s of illumination and 3 s in darkness for every cycle.

formation that causes photodegradation[47]. For instance, depassivated iodide ions will act as reservoirs for binding sites to form $I^0$ species, which can easily combine to generate $I_2$, leading to defect formation to compromise device performance[48]. To verify the susceptibility to defect formation following the depassivation, we fabricated inverted solar cells with the configuration ITO/4PADCB/$Al_2O_3$/$Cs_{0.05}FA_{0.85}$$MA_{0.1}PbI_3$/LiF/$C_{60}$/BCP/Ag, adopting the high-efficiency HP formula $Cs_{0.05}FA_{0.85}MA_{0.1}PbI_3$ widely used in the community[49]. As shown in Fig. 4D, the pristine solar cell without any ammonium-based passivation exhibited a champion PCE of 17.23% (open-circuit voltage $V_{oc}$ = 1.10 V, short-circuit current $J_{sc}$ = 22.09 mA/cm², fill factor FF = 70.71%). In contrast, proper passivation of the HP layer using PPA and octane-1,8-diammonium significantly improved the device performance, resulting in a champion PCE of 26.44% ($V_{oc}$ = 1.179 V, $J_{sc}$ = 26.22 mA/cm², FF = 85.49%) in our lab. The certified PCE by a third party is 26.15% (for reverse scan, $V_{oc}$ = 1.188 V, $J_{sc}$ = 25.86 mA/cm², FF = 85.12%, Fig. S35). This contrast confirms the beneficial passivation effect induced by the ammonium ligands. Part of the improvement also arises from the crystallization modulation effect of the ammonium ligands[50].

The harm caused by violet/UV photons to HP solar cells can occur in several ways. To isolate the effect of the depassivation mechanism here, we subjected the passivated solar cells to a fatigue test with intermittent light-dark alternation. A detailed discussion can be found in Supplementary Note 5. During each replication, the solar cell was first illuminated for an I–V scan for 7 s, and then allowed to recover in

the dark for 3 s for the self-healing of metastable states[51]. Illumination will evoke a defect-annihilation process ($I_i^-$-$V_I^+$ Frenkel pairs annihilation) and a defect generation process (two $I_i^0$ combining into $I_2$), and the defect generation process tends to occur in a defective region where $I^-$ is poorly stabilized[48,52]. The built-in potentials of the heterojunctions vary during the I-V scan in each replication, causing periodic changes in the built-in electric field that affect the ions in the depletion region (Fig. S36).

We used two illumination conditions for comparison: violet/UV light-filtered solar illumination (using a > 420 nm long-pass filter) and unfiltered illumination. The solar simulator we used was measured to contain 12.78% of its total irradiance at wavelengths below 420 nm, slightly higher than that in the standard AM 1.5 G irradiation (~5%). The filtered light intensity was calibrated to compensate for the filtering loss. Besides, to exclude interference from ambient factors, we conducted the tests in an $N_2$-filled glovebox. As shown in Fig. 4E, for both illumination conditions, the PCEs increase after the initial several replications, indicating the dominance of the defect annihilation process initially[48,52]. The device subjected to the unfiltered illumination manifests a weaker PCE increase, which implies a less favored defect annihilation process. This result is consistent with the violet/UV light-induced depassivation mechanism, i.e., the ligand anchoring for $I^-$ sites is disrupted, so the defect generation process is exacerbated. Indeed, the absorption spectrum characterization throughout this test reveals a significantly more pronounced decline in light absorption for the unfiltered device (Fig. S37), which can be ascribed to more pronounced

defect generation. Subsequently, the PCE starts to decline as the defect formation process gradually takes over. The decline happens first in the unfiltered device. After 47 replications, the device under the filtered illumination retains 98.0% of its initial PCE, while the value significantly drops to 91.4% for the unfiltered device. In another test protocol, after being subjected to continuous illumination for 24 h, the unfiltered device exhibits a more severe decline in $V_{oc}$ and FF compared to the filtered device (Fig. S38, Table S1), a pattern indicating defect formation as $V_{oc}$ and FF are sensitive to defects (Supplementary Note 6). These results show that the light-disrupted ammonium-HP interaction mechanism depassivates HP to make it susceptible to defect formation under illumination, causing photodegradation.

## Discussion

We believe the mechanism revealed here could also contribute to the understanding of many other important phenomena in this field, such as the dynamism of the 2D/3D HP interface[13,53,54], phase segregation in mixed-halide HPs[4], ion migration and hysteresis in devices[55], enhancements in photocatalytic activity[43], and the photoluminescence blinking/quenching of HP nanocrystals[56]. Specifically, the 2D/3D HP configuration, wherein a 2D HP layer serves as a protective and passivating coating over a 3D HP absorber, exhibits a dynamic dimensionality change at the interface over time[54]. This dynamism causes deviations from the initial design and compromises device efficiency and stability. This dynamic behavior leads to deviations from the initial structural design, compromising both device efficiency and stability. Previous studies have observed the migration of ammonium ligands under illumination, consistent with the mechanism proposed herein[57]; Ion migration, a prevalent phenomenon in HPs, underpins critical device behaviors such as hysteresis[55]. While anchoring ammonium ligands can mitigate ion migration, this effect may be exacerbated under illumination[58]. This photoresponse aligns with the proposed mechanism; For photocatalytic applications of 2D HPs, such as $H_2$ evolution[43], efficient separation of electrons and holes is essential. Our mechanism suggests that this separation is facilitated by an organic-to-inorganic carrier transition induced by violet/UV photons, thereby enhancing photocatalytic activity. Additionally, ammonium ligands are commonly employed as surface-passivating agents for HP nanocrystals. However, a light-induced depassivation mechanism may trigger ligand dynamics at the surface, exposing active sites and introducing trap states. This process could contribute to charge trapping/detrapping phenomena, resulting in photoluminescence blinking or quenching[56]. Nevertheless, further experimental investigations are required to substantiate these hypotheses.

The depassivation mechanism works because violet/UV photons are sufficient to pump the carrier transition from the ligand HOMO to the HP $E_x$/CBM. Therefore, it should be prevented if the HOMO-CBM gap increases beyond the violet-UV photon energy. Towards this, we believe the following directions might merit consideration:

1) Ammonium ligands exhibit adjustable chemical properties. Functional groups on these ligands can be engineered to increase the ionization potential of the ammonium moiety.
2) Given that this mechanism creates a tendency to disrupt the interaction between ammonium ligands and the HP lattice, multi-site anchoring should be conducive to alleviating the light-induced depassivation.
3) Instead of using ammonium ligands directly as passivating agents, 2D HPs can be a sturdier alternative. 2D HPs can not only be tailored in the ammonium ligands but also in the HP sublattice. By decreasing the electron affinity of the HP lattice (e.g., by using non-lead 2D HPs or alloying), the HOMO-$E_x$/CBM gap can also be increased.
4) New types of ligands beyond ammonium-terminated ligands should be developed for HP solar cells. For instance, phosphorus-, oxygen-, and sulfur-centered ligands, such as phosphonium salts,

carboxylic acids, and thiols, show great promise. However, their effect in passivation and crystallization control is not as prevalent as that of ammonium ligands, necessitating further advances.

To conclude, our findings reveal a universal mechanism by which violet/UV light causes the depassivation of ammonium-terminated ligands, a leading class of defect-passivating agents for HP cells. Specifically, violet/UV photons trigger an electronic transition from the ligands to the HP lattice, causing the deprotonation of these ligands. This deprotonation disrupts the ligand-HP interaction and thus gives rise to depassivation. This mechanism applies whether the ammonium-terminated ligands act directly as passivating agents or generate 2D HPs for passivation. These findings provide crucial insights into the photodegradation of state-of-the-art HP solar cells despite meticulous passivation.

## Methods

### Materials

The materials used in this study include lead(II) iodide ($PbI_2$, >99.999%); cesium iodide (CsI, 99.999%); methylammonium iodide (MAI, 99.5%); methylammonium hydrochloride (MACl, 99.5%); butylammonium iodide (BAI, 99.0%); octylammonium iodide (HAI, 99.5%); dodecylammonium iodide (DDAI, 99.5%); phenethylammonium iodide (PEAI, 99.5%); phenylpropylammonium iodide (PPAI, 99%); octane-1,8-diammonium iodide (ODADI, 99.5%); lead(II) bromide ($PbBr_2$, 99.99%); methylammonium lead iodide ($MAPbI_3$, 99%); $C_{60}$ (99%); bathocuproine (BCP, 99%); and phenylpropylammonium bromide (PPABr, 99%), which were purchased from Xi'an Polymer Light Technology Corporation.

Additional materials including lithium fluoride (LiF, 99.995%); dimethylformamide (DMF, anhydrous, 99.8%); dimethyl sulfoxide (DMSO, anhydrous, $\geq$ 99.9%); octadecylammonium (ODA, $\geq$ 99% (GC)); chlorobenzene (CB, anhydrous, 99.8%); ethyl alcohol (anhydrous, $\geq$ 99.5%); acetone ($\geq$ 99.5%); isopropanol (IPA, anhydrous, 99.5%), and $Al_2O_3$ dispersion (nanoparticles, < 50 nm particle size (DLS), 20 wt. % in isopropanol) were sourced from Sigma-Aldrich.

Formamidinium iodide (FAI, > 99.99%) was purchased from Greatcell Solar Material. 4PADCB was purchased from Suzhou LiWei Tech Co., Ltd. Acetic acid (HAc, $\geq$ 99.8% (GC)) was sourced from Aladdin.

Unless otherwise stated, these reagents were used directly without further purification.

### 2D HP film fabrication

The ITO substrates were sequentially cleaned using an ultrasonic bath. Each cleaning step consisted of a detergent solution, deionized water, acetone, and IPA, with each step lasting 15 min. After cleaning, the substrates were placed in a drying oven to ensure complete evaporation of all solvents. Before spin coating, the substrates received a 15-min UV treatment.

For the preparation of the $BA_2PbI_4$ film, the precursor was created by mixing $PbI_2$ and BAI in a molar ratio of 1:2, dissolving the mixture in a solvent blend of DMF and DMSO at a volume ratio of 7:3. The lead(II) concentration in the precursor was set to 1 M for thick films and 0.1 M for thin films. The precursor solution was then deposited onto the substrate using spin coating. Specifically, 50 μL of the precursor was spin-coated onto the substrate at a speed of 4000 rpm for 30 s, with a ramp-up rate of 2000 rpm per second. Following spin coating, the film was annealed at 100 °C for 10 min. All fabrication procedures were conducted within a nitrogen-filled glovebox.

For the synthesis of $HA_2PbI_4$, $OA_2PbI_4$, $DA_2PbI_4$, $PEA_2PbI_4$, and $PPA_2PbI_4$, the preparation of the precursor and film fabrication procedures was identical, with the only variation being the specific ammonium iodide reagents used in the precursor solutions, tailored to the target 2D HP type.

## ITO/2D HP/Cu device fabrication for the steady-state optoelectrical characterizations

The cleaning of substrates, the preparation of various 2D HP precursors, and the spin coating parameters for the 2D HP film fabrication are the same as above. After obtaining the 2D HP layer, a 100 nm-thick Cu electrode was thermally deposited onto the 2D HP layer to complete the device.

## 3D HP (MAPbI3) film fabrication

The precursor was prepared by dissolving stoichiometric amounts of $PbI_2$ (461.01 mg, 1 mmol) and MAI (158.97 mg, 1 mmol) in 1 mL of DMF. The substrate cleaning procedure was consistent with the previous steps. The film fabrication employed a two-step spin coating process, first at a speed of 1000 rpm for 10 s and then at 5000 rpm for 30 s. The ramp-up rate for both steps was 2000 rmp/s. For each spin coating iteration, 40 μL of the precursor was used. Additionally, 800 μL of CB was deposited onto the wet HP film as an antisolvent at the $15^{th}$ second during the second spin coating step. Following spin coating, the film was annealed at 100 °C for 10 min. All these fabrication procedures were carried out within a nitrogen-filled glovebox.

## Synthesis of ODA2PbI4 nanoflakes for the UV light-driven photoreaction

The precursor solution was prepared in two steps. First, CsI (519.62 mg, 2 mmol) and $PbI_2$ (461.01 mg, 1 mmol) were dissolved in 15 mL of DMSO to create precursor A. Separately, 0.5 g of ODA was dissolved in 10 mL of HAc to create precursor B.

Under vigorous stirring, 1.5 mL of precursor A was injected into 10 mL of precursor B. This triggered an instantaneous color change of the transparent solution to yellow due to the formation of the $ODA_2PbI_4$ nanoflakes. The resulting solution was then subjected to 15 min of ultrasonic treatment.

After that, the $ODA_2PbI_4$ precipitation was extracted via centrifugation. The precipitate was then washed twice with HAc and once with toluene.

All these preparations and purification procedures were carried out in the ambient air.

## Theoretical calculations

All first-principle calculations in this work were carried out by using the Vienna Ab initio simulation package (VASP) based on density DFT with the projector augmented wave (PAW) method[59–61]. The generalized gradient approximation (GGA) of Perdew–Burke–Ernzerhof (PBE) was used for the exchange-correlation function[62]. A cutoff energy of 500 eV for the plane-wave basis set is adopted in all computations. Each unit cell with a $5 \times 5 \times 1$ Γ centered k-point grid was employed for geometry optimization and energy calculations. More details are presented in Fig. S23, Supplementary Note 1.

## UV light-driven photoreactions

UV light-driven photoreactions were conducted in a custom-designed double-layered Pyrex vessel. The HI aqueous solution was prepared by mixing 16 mL of HI, 4 mL of $H_3PO_2$, and 20 mL of deionized water. Next, 30 mg of $ODA_2PbI_4$ powder was added to the HI solution, which was then subjected to ultrasonic treatment for 30 min before transferring to the vessel.

To ensure accurate hydrogen production and eliminate interference from $O_2$ in the air, the reaction system was vacuumed prior to irradiation. A Xe lamp (Ushio−CERMAX LX300) served as the light source, with the light intensity fixed at 250 mW cm⁻². The irradiation spectrum was controlled using a 420 nm long-pass filter and an AM 1.5 G filter. The amount of evolved $H_2$ was measured using gas chromatography. During the test, the temperature of the vessel was fixed at 10 degrees Celsius to exclude the interference from heat.

## HP solar cells fabrication

The solar cell device structure was configured as follows: ITO/4PADCB/$Al_2O_3$/$Cs_{0.05}FA_{0.85}MA_{0.1}PbI_3$/LiF/$C_{60}$/BCP/Ag. Before thin-film deposition, the ITO substrates underwent a thorough cleaning process as described earlier.

For the hole-transport layer, 4PADBC was dissolved in ethyl alcohol to a concentration of 0.5 mg/ml. The solution was dropped onto the substrate and left to stand for 10 s. Then, the solution was spin-coated at 3000 rpm for 30 s, and the ramp-up rate was 2000 rpm/s. After that, the coated layer was annealed at 100 °C for 10 min. Following the 4PADBC layer, $Al_2O_3$ nanoparticles were deposited to increase hydrophilicity. The $Al_2O_3$ dispersion was obtained by diluting the purchased one 50 times with IPA. The dispersion was vigorously shaken before use. It was deposited onto the 4PADBC layer by spin-coating at 5000 rpm for 30 s, and the ramp-up rate was 3000 rpm/s. After the deposition, the film was annealed at 100 °C for 10 min.

The HP precursor was prepared by dissolving the following materials in a mixture of DMF (0.8 mL) and DMSO (0.2 mL): $PbI_2$ (760.70 mg), FAI (219.30 mg), MAI (23.80 mg), CsI (19.5 mg), and MACl (12.70 mg). ODADI (0.48 mg) was also added as a bulk passivating agent. The HP precursor solution (50 μL) was then spin-coated onto the substrate at 4000 rpm for 40 s, with a ramp-up rate of 800 rpm/s. At the $35^{th}$ second of the spin-coating process, 150 μL of CB was deposited onto the substrate as an antisolvent. Subsequently, the ITO/4PADCB/$Al_2O_3$/$Cs_{0.05}FA_{0.85}MA_{0.1}PbI_3$ substrate was annealed at 100 °C for 30 min. In addition to the bulk passivation, surface passivation was also applied. Specifically, a PPABr/IPA solution (0.5 mg mL⁻¹, 30 μL) was dynamically spin-coated onto the HP film at 4000 rpm for 30 s, with a ramp-up rate of 2000 rpm/s. Finally, the film was annealed at 100 °C for 5 min.

Subsequently, a series of thin films was thermally evaporated under high vacuum conditions to complete the device structure. First, a 1 nm-thick layer of LiF was deposited, followed by a 24 nm-thick layer of $C_{60}$, and then a 4.5 nm-thick layer of BCP. Finally, a 100 nm-thick Ag electrode was thermally deposited to finalize the device fabrication.

## FTIR spectra measurement of UV light-irradiated and non-irradiated 3D MAPbI3 (with ammonium passivation)

100 mg MAPbI3 powders were dispersed in 5 mL CB, and 0.162 mmol BA/oleic acid pairs were used as ligands. The target sample was subjected to UV irradiation (365 nm @-25 mW/cm²) and gently stirred with a magnetic stirrer at the same time. The reference sample was treated the same way, except it was encapsulated with aluminum foil to prevent irradiation. The irradiation test lasted for 5 h. After every hour, part of MAPbI3 was extracted and isolated by centrifugation and vacuum treatment. Finally, the irradiated MAPbI3 sample, the non-irradiated MAPbI3 sample, and the pristine MAPbI3 were sent for the FTIR spectroscopy characterization.

## Characterizations

Time-resolved photoluminescence (TRPL) characterizations for 2D HP films were conducted with a universal streak camera (Hamamatsu). Two excitations of 2.76 eV (1 kHz, 100 fs, for excitonic photoexcitation) and 3.82 eV (1 kHz, 100 fs, for the ammonium-to-HP carrier transition) were used for comparison, with a similar injected carrier density of ~$2 \times 10^{-12}$ cm⁻²;

Steady-state UV−vis absorption spectra were obtained from a UV−vis/NIR spectrophotometer (JASCO V-770EX);

The TA characterizations were conducted using a HELIOS TA spectrometer. An excitation light of 3.82 eV (325 nm, 1 kHz, 100 fs) was used for 2D HP films, and the power was 0.318-2.29 mW/cm² (recorded in the figure caption for each sample);

The steady-state PL spectra were obtained by a Cary Eclipse Fluorescence Spectrophotometer.

The XRD measurements were carried out with a Bruker D8 Advance XRD system and an Ultima IV (Rigaku) system.

The scanning electron microscope (SEM) images were obtained by an FEI field emission electron microscope, Quanta 250 F.

The UPS and X-ray Photoelectron Spectroscopy (XPS) measurements were implemented using an electron analyzer (ESCALAB 250Xi, Thermo Fisher Scientific), which is equipped with a UV source (hυ = 21.2 eV) for the UPS measurement and a monochromatic X-ray source (hυ = 1486.7 eV) for the XPS measurement. The UPS and XPS measurements were carried out in a vacuum ($1.0 \times 10^{-10}$ mbar). For the UPS characterization, a bias voltage of −5.0 V was applied.

J–V curves were extracted under AM 1.5 G illumination, enabled by a solar simulator (Newport IEC/JIS/ASTM) equipped with a 450 W xenon lamp and a Keithley 2400 source meter. The intensity of the simulated solar light was standardized by a standard Si photodiode detector calibrated at the National Renewable Energy Laboratory. The Keithley 2400 source measure unit was used to record the device parameters.

## Reporting summary

Further information on research design is available in the Nature Portfolio Reporting Summary linked to this article.

## Data availability

The data supporting the findings generated in this study have been deposited in the Figshare database and are accessible under the identifier https://doi.org/10.6084/m9.figshare.30058753.v1. https://figshare.com/articles/dataset/Violet_ultraviolet_light-induced_depassivation_in_halide_perovskite_solar_cells/30058753.

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

## Acknowledgments

This work is funded by the Science and Technology Development Fund, Macao SAR (File no. 0010/2022/AMJ, 0060/2023/RIA1, 0046/2025/RIB1, 0136/2022/A3, 0122/2024/AMJ, 0002/2024/TFP), UM's research fund (File no. MYRG-GRG2023-00065-IAPME-UMDF, MYRG-GRG2024-00156-IAPME, MYRG-CRG2025-0024-IAPME, MYRG-GRG2025-00259-IAPME), and the Natural Science Foundation of China (62175268, 62288102, 22405010) to GCX; and by Shenzhen Science and Technology Program (RCJC20200714114434086), and Research Team Cultivation Program of Shenzhen University (2023QNT013) by CLS; and by the Science and Technology Development Fund, Macao SAR (File no. 0042/2025/RIA1), the Science and Technology Development Fund, Macao SAR (File no. 0148/2024/RIA2), and China Postdoctoral Science Foundation (No. 2021M702227) to DJY.

## Author contributions

The study was conceptualized by D.J.Y., F.C., G.C.X., T.W.L., C.L.S., and J.W.Z. The methodology was developed by F.C., D.J.Y., X.S.Q., Y.C., G.W., Z.P.Z., and Y.L.M. Investigation was conducted by F.C. and D.J.Y., while visualization of the results was handled by F.C., D.J.Y., X.S.Q., and Y.C. Funding acquisition was secured by G.C.X., J.W.Z., C.L.S., and D.J.Y., with project administration overseen by G.C.X., C.L.S., J.W.Z., and D.J.Y. Supervision was provided by G.C.X., T.W.L., C.L.S., J.W.Z., and W.H. The original draft of the manuscript was written by F.C. and D.J.Y., and the writing, review, and editing process involved contributions from D.J.Y., F.C., G.C.X., T.W.L., X.S.Q., and Y.C.

## Competing interests

The authors declare no competing interests.
