## [Transparent Peer Review file · Nature Communications]

Violet/ultraviolet light-induced depassivation in halide perovskite solar cells

Corresponding Author: Professor Guichuan Xing

Version 0:

Reviewer comments:

Reviewer #1

(Remarks to the Author)

This paper reveals for the first time the mechanism of ammonium-terminated ligand depassivation induced by violet/ultraviolet light, filling the previous research gap in light-induced ligand failure. The authors clarify the electron transfer from ammonium-terminated ligands to perovskite lattices under UV excitation and the subsequent deprotonation process, providing new insights into understanding light degradation in high-efficiency perovskite solar cells. However, several issues require clarification and revision. Please address the following questions before acceptance:

1. The introduction mentions existing studies only focus on thermal instability and iodine oxidation, but fails to further compare the relative contributions of violet/ ultraviolet light-induced failure with other mechanisms (e.g., thermal stress). This weakens the necessity justification for this research.
2. In Figure 2, the ΔV differences between thick/thin films demonstrated through I-V curves are used as evidence for light-induced interfacial dipoles. Could electrode contact differences or uneven distribution of photogenerated carriers interfere with ΔV ? For instance, limited light penetration depth in thick films might cause carrier accumulation near electrode regions.
3. While the conclusion briefly summarizes the UV-induced de-passivation mechanism of ammonium ligands and mentions its potential implications for other phenomena (e.g., ion migration, phase separation), these discussions lack in-depth exploration and data support in the main text, appearing vague.
4. The photocatalytic hydrogen production experiment (Figure 3) indirectly speculates deprotonation of ammonium ligands through H₂ monitoring. Why weren't direct characterizations (e.g., XPS or NMR) to confirm the chemical conversion from -NH₃⁺ to -NH₂?
5. What scientific question does the absorption spectrum data in Figure 4B aim to address? Please elaborate on this section.
6. In Figure 4C, the weakened ligand signals in FTIR alone are insufficient to convincingly prove the disruption of ammonium-HP interactions and attribute this process to ligand deprotonation. Please provide further justification.

Reviewer #2

(Remarks to the Author)

Research by Yu et al. discovered that violet/ultraviolet light can induce depassivation of ammonium -terminated ligands. These ligands become deprotonated, thereby leading to the depassivation phenomenon. This provides a novel insight into the violet/ultraviolet light-induced degradation of highly efficient perovskite solar cells prepared with meticulous passivation, aiding in understanding the dynamic behavior at the two-dimensional/three-dimensional efficient photovoltaic interface. However, the experimental evidence for some key conclusions still needs to be strengthened, especially in terms of the universality of the mechanism, the characterization of dynamic processes and the influence of environmental factors. While the perspective offered by this research work is novel, several points require clarification, as detailed below:

1. The authors claim that violet/ultraviolet light induces the deprotonation of amines, and this deprotonation disrupts the ligand-perovskite interaction, triggering the depassivation of the perovskite making it susceptible to light degradation. However, the issue is that UV light itself can induce the degradation of perovskite, regardless of whether it is passivated or not. Furthermore, UV light also damages electron/hole transport layer materials. The reduction in perovskite solar cell performance is the result of the combined effects of UV light.
2. When measuring the device IV curves under filtered/non-filtered light conditions, the performance decline in non-filtered devices is primarily due to reductions in VOC and FF, while JSC is less affected. Why is this? Correspondingly, the authors

show in Figure 4D that the short-circuit current density of the unpassivated device is only 22.09 mA/cm², while that of the passivated device significantly increases to 26.22 mA/cm². How do the authors explain this?

3. The authors propose the phenomenon of violet/ultraviolet light-induced depassivation in passivated perovskite solar cells but do not provide specific methods to address this problem in order to obtain UV-stable perovskite solar cells.

4. It would be beneficial if the authors could provide the EQE of the perovskite solar cells before and after UV aging, along with corresponding analysis, to gain a deeper understanding of the perovskite's photo-response.

5. From the data in Figure 4E, it can be seen that under illumination of 1.064 suns, the performance of the perovskite solar cell decreased by nearly 10% in less than 300 seconds of light exposure. This cannot simply be attributed to ultraviolet light in the solar spectrum; the perovskite cell itself has severe intrinsic light stability issues. Even under UV-filtered light, the performance of the perovskite cell also showed a significant decrease within the same timeframe, dropping from a peak of about 1.05 to around 0.98.

6. Another intriguing phenomenon: during the light/dark cycling test, the efficiency of the perovskite solar cells increased significantly under both UV-filtered and UV-retained light, but the increase was greater under UV-filtered light. Why is this? Additionally, what was the initial efficiency of the devices used for the light/dark cycling test?

7. It is mentioned in the text that the depassivation mechanism induced by violet/ultraviolet light is generally applicable to ammonium-based ligands, but currently, near-ultraviolet light bleaching (PB) signals have only been observed in a few 2D perovskites and quasi-2D systems such as BA₂PbI₄. It is suggested to supplement more 3D perovskite data of ammonium ligands with different structures (such as those containing aromatic rings, long-chain alkyl groups, etc.), especially the direct evidence of electron transfer at the ligand-perovskite interface in the pure 3D MAPbI₃ system, to confirm the universality of the mechanism in different dimensions and ligand structures.

8. The near-ultraviolet PB signal is attributed to the electron transfer from ammonium to perovskite, but the influence of impurities (such as unreacted PbI₂ or ligand decomposition products) is not completely excluded in the manuscript. Although XRD shows that the purity of some samples is relatively high, it is necessary to further confirm through characterization methods such as high-resolution transmission electron microscopy (HRTEM) or time-of-flight secondary ion mass spectrometry (TOF-SIMS) that the signals only come from the ligand-perovskite interface, rather than residual precursors or degradation products.

9. In the manuscript, the HOMO/LUMO energy levels of ligands are defined based on the DFT calculation results of Silver et al. However, different calculation methods (such as functional selection and basis set size) may lead to deviations in energy values. It is suggested to supplement the first-principles calculations for specific ligands (such as BA⁺), clarify the consistency degree between the energy difference of HOMO-exciton (Ex) and the energy of the near-ultraviolet PB signal observed experimentally, and discuss the possible sources of the deviation between the calculation and the experiment (such as solvent effect, exciton binding energy).

10. Figure 4E shows that the device efficiency decreases under ultraviolet light, but the current data only infer the formation of defects through the I-V curve. It is suggested to supplement in-situ characterization methods, such as time-resolved photoluminescence (TRPL) under light or spatially resolved surface photovoltage spectroscopy (SPS), to directly monitor the density changes of non-radiative recombination centers caused by ligand depassivation, and establish the real-time correlation between photocurrent attenuation and ammonium dissociation.

11. In the light reaction experiment, the increase in the generation rate of H₂ indicates the deprotonation of the ammonium group, but there is a lack of kinetic curves under different light intensities (such as the variation of H₂ generation amount with time). It is suggested to supplement the ultraviolet light intensity gradient experiment to quantify the relationship between photon flux and deprotonation rate, and analyze the activation energy of this process through the Arrhenius equation to distinguish light-induced electron transfer from thermally driven ligand decomposition.

12. Fatigue tests were conducted in a nitrogen glove box, but in practical applications, water vapor and oxygen may accelerate the degradation of perovskite. It is suggested to supplement the comparative experiments in the air environment to clarify whether the depassivation induced by violet/ultraviolet light is independent of environmental factors or has a synergistic effect with water vapor. Furthermore, it is necessary to explain whether the intensity of the ultraviolet light used in the test matches the ultraviolet component in the actual sunlight (for example, the ultraviolet proportion in the AM 1.5G spectrum is approximately 5%).

13. The FTIR results show that the ligand signal weakens after ultraviolet irradiation, but it is only a static characterization. It is recommended to use in situ FTIR to monitor the real-time changes of ammonium characteristic peaks (such as N-H scaling vibration) during the irradiation process, clarify the dynamic path and reversibility of deprotonation (-NH₃⁺ → -NH₂), and rule out the possibility of physical desorption of ligands.

Version 1:

Reviewer comments:

Reviewer #1

(Remarks to the Author)

The paper can be published now since all the questions have been well addressed though the the degradation interference from other mechanisms were not separated.

Reviewer #2

(Remarks to the Author)

The authors have responded the proposed questions. This manuscript can be accepted now.

Catalogue

Response to Reviewer #1:.....	2
Comment #1	2
Comment #2	3
Comment #3	9
Comment #4	11
Comment #5	14
Comment #6	16
Response to Reviewer #2:.....	20
Comment #1	20
Comment #2	23
Comment #3	27
Comment #4	28
Comment #5	31
Comment #6	33
Comment #7	35
Comment #8	40
Comment #9	43
Comment #10	46
Comment #11	49
Comment #12	52
Comment #13	55

Response to Reviewer #1:

This paper reveals for the first time the mechanism of ammonium-terminated ligand depassivation induced by violet/ultraviolet light, filling the previous research gap in light-induced ligand failure. The authors clarify the electron transfer from ammonium-terminated ligands to perovskite lattices under UV excitation and the subsequent deprotonation process, providing new insights into understanding light degradation in high-efficiency perovskite solar cells. However, several issues require clarification and revision. Please address the following questions before acceptance:

Comment #1. The introduction mentions existing studies only focus on thermal instability and iodine oxidation, but fails to further compare the relative contributions of violet/ultraviolet light-induced failure with other mechanisms (e.g., thermal stress). This weakens the necessity justification for this research.

Response: We sincerely thank you for dedicating your valuable time to reviewing our manuscript and for providing insightful comments. We have carefully revised the manuscript to address your feedback to the best of our ability. We hope the refined version more effectively conveys our scientific findings and contributes meaningfully to the perovskite research community.

We agree that the justification in the introduction needs to be enhanced. While we agree that there are many degradation paths in halide perovskites, including ultraviolet light-triggered ones, we would like to clarify that we would like to focus on the passivating failure aspect, which is a different perspective from general degradation.

As we discussed in the manuscript, the current understanding of the passivation failure of ammonium ligands, the most widely used passivating agents in halide perovskites, is limited to iodide oxidation (in the precursor stage) and thermal/light intolerance (in formed perovskite films). To the best of our knowledge and with your kind agreement, our work represents the first report on ultraviolet/violet light-induced depassivation in halide perovskites.

To strengthen the discussion and argument, we have made revisions in the revised

manuscript (on page 4) following your comment. The revisions are highlighted in yellow, as follows:

“However, contradictory to the excellent defect passivation by ammonium ligands, rapid degradation during operation remains a critical challenge for these state-of-the-art HP cells, which raises concerns about the durability of ammonium ligands. Increasing attention is paid to studying the depassivation of ammonium ligands. Meng *et al.* showed that deprotonation of organic cations can occur in the precursor stage with the oxidation of iodide ions.^[28] Wang *et al.* reported that ammonium ligands with a low acid dissociation constant can be thermal-intolerant and might fail passivation under high-temperature light soaking.^[29] The poor thermal/light tolerance of typical ammonium ligands was also confirmed by Pei *et al.*^[30] However, the depassivation mechanisms of ammonium ligands are still insufficiently understood and warrant further exploration.”

References

28. Meng, H. *et al.* Inhibition of halide oxidation and deprotonation of organic cations with dimethylammonium formate for air-processed p–i–n perovskite solar cells. *Nat. Energy* **9**, 536-547 (2024).
29. Wang, M. *et al.* Ammonium cations with high pKa in perovskite solar cells for improved high-temperature photostability. *Nat. Energy* **8**, 1229-1239 (2023).
30. Pei, F. *et al.* Inhibiting defect passivation failure in perovskite for perovskite/Cu(In, Ga)Se₂ monolithic tandem solar cells with certified efficiency 27.35%. *Nat. Energy* **10**, 824-835 (2025).

We sincerely hope the above reply addresses your concern.

Comment #2. In Figure 2, the ΔV differences between thick/thin films demonstrated through I-V curves are used as evidence for light-induced interfacial dipoles. Could electrode contact differences or uneven distribution of photogenerated carriers interfere with ΔV ? For instance, limited light penetration depth in thick films might cause carrier accumulation near electrode regions.

Response: Thank you for your insightful comment. In our case, electrode contact differences or uneven distribution of photogenerated carriers do exist, but they do not affect the interpretation of the light-induced global potential difference (ΔV).

FIRST, in terms of the potential impact of electrode contact differences, we used an ITO/2D perovskite/Cu MSM device configuration for the measurement. ITO and Cu are selected due to their similar work functions (ITO in the range of 4.6-4.75 eV and Cu at 4.7 eV), minimizing the interference from the contact difference.

Figure R1. I-V characteristic of the ITO/thick BA_2PbI_4 /Cu device, which shows an Ohmic contact for both metal/semiconductor contacts on the two sides.

All devices based on different types of 2D perovskites show an Ohmic feature. Take BA_2PbI_4 (BA denotes butylammonium) as an example, the I-V characteristic measured in the dark reveals an Ohmic feature, as shown in Figure R1 (presented as Fig. S13 in the prior manuscript). The I-V characteristics for other 2D perovskite-based devices that also show an Ohmic feature can be found in Fig. S18 -S22 in the Supplementary Information file. **Notably, the Ohmic contact feature indicates that the Schottky barrier heights (from the BA_2PbI_4 side to the metal electrode side) for the two metal/semiconductor contacts are minor.** Otherwise, a Schottky contact feature

should be observed. Additionally, the I-V characteristic measured under standard AM 1.5G illumination **shows no photovoltaic effect** (no photovoltage), which also confirms a negligible impact of the electrode Fermi level difference, as a high disparity in the work functions of the two electrodes should induce a photovoltaic effect. Therefore, the impact of electrode contact difference should be trivial. Further clarification is embedded in the following discussion of the uneven distribution of photogenerated carriers.

Figure R2. I-V characteristics of the thick BA_2PbI_4 film in the dark, under full-spectrum one-sun illumination, and under one-sun illumination excluding violet/UV photons (energy > 2.95 eV), measured in the device configuration of ITO/ BA_2PbI_4 /Cu.

SECOND, in terms of the potential impact of uneven distribution of photogenerated carriers, we believe the uneven distribution exists and will impact the energy landscape across the device, but it will not interfere with the interpretation of the result.

Figure R2 shows the I-V symmetry characteristic of the ITO/thick BA_2PbI_4 /Cu device. When in the dark, the $(I_{for} + I_{rev})$ versus V_{abs} characteristic shows a mixed exponential/linear feature, as the exponential fit (single exponential is the best fit) shows a non-ideal R^2 value of 0.95616. The exponential feature corresponds to a Schottky contact, while the linear feature corresponds to an Ohmic contact. The imperfect exponential fit agrees with the dominant Ohmic contact feature present in

Figure R1, and there is no dramatic band bending at the contacts to cause Schottky contact properties.

When the device is under full-spectrum simulated AM 1.5G illumination, the $(I_{\text{for}} + I_{\text{rev}})$ versus V_{abs} dependence reverses, and the $(I_{\text{for}} + I_{\text{rev}})$ instead increases with V_{abs} . This dependence can be well single-exponentially fitted with an R^2 value of 0.99295, which suggests the emergence of an inverse ΔV upon illumination. This result is consistent with the effect of the interfacial dipole arising from the light-induced ammonium-to-perovskite carrier transition in our mechanism. To further validate that this phenomenon is due to violet/ultraviolet photons, we used a long-pass filter that filters out photons with a wavelength shorter than 420 nm. Indeed, the I-V characteristic becomes almost flat with a non-ideal exponential fit ($R^2 = 0.93034$), suggesting significantly mitigation of ΔV . This change indicates that the emergence of ΔV is indeed due to violet/ultraviolet photons, in line with our mechanism.

Figure R3. UPS spectrum of the BA_2PbI_4 film measured under weak illumination. The Fermi level is about -4.6 eV versus the vacuum level, which is close to the center of the bandgap. The CBM of BA_2PbI_4 is -3.4 eV, and the VBM is -5.8 eV.

Notably, the I-V characteristic under the filtered illumination does not revert to

that in the dark. We believe this is where the uneven charge carrier distribution takes effect. That is, the photoinjected electron-hole pairs (excitons in 2D perovskites) would concentrate near the illuminated side due to limited penetration depth, and they would screen the built-in potential, despite being weak, in the ITO/BA₂PbI₄ junction. To confirm this claim, we have carried out ultraviolet photoelectron spectrum (UPS) characterization of the BA₂PbI₄ film in the dark and under illumination. The illumination source in this UPS measurement is broad-spectrum visible white light, covering approximately 400-700 nm, with a color temperature of around 3000-5000 K, and it is significantly less intense compared to the simulated AM 1.5G illumination (~0.1% intensity suggested by online data). Even with this low intensity illumination, the Fermi level of BA₂PbI₄ shifts to the middle of the bandgap upon illumination (Figure R3), which means the photoinjected carrier density is already enough to fully screen the built-in potential near the illumination side.

Importantly, even with this screening effect due to uneven charge carrier distribution, the interpretation of the I-V characteristic is still valid to support our mechanism. The energy landscape across the device when in the dark and under different illumination conditions can be illustrated as below in Figure R4:

Figure R4. Schematic illustration of the energy landscapes across the device when in the dark and under different illumination conditions.

When **in the dark**, the ITO/BA₂PbI₄ and BA₂PbI₄/Cu contacts form weak built-in potentials, but the Ohmic property is dominant. The minor work function difference between ITO and Cu causes non-perfect symmetry across the energy landscape, as illustrated in Figure R4a. The downward band bending is based on the weak p-type property of BA₂PbI₄ (Fermi level at 4.74 eV versus the vacuum level, closer to VBM, shown in Figure S11 in the manuscript). When the device is under **filtered simulated AM 1.5G illumination**, in which photons with a wavelength shorter than 420 nm are filtered out, uneven charge carrier concentration occurs due to limited penetration depth of the lights (and also limited diffusion length of excitons, referring to Magdaleno, A. J. et al. *Mater. Horiz.* 8, 639-644 (2021)). Consequently, the built-in potential at the ITO/BA₂PbI₄ contact is screened, while that of the BA₂PbI₄/Cu contact on the back will be less impacted. The corresponding energy landscape is illustrated in Figure R4b. When the device is **under full-spectrum illumination that maintains the <420 nm photons**, the energy barrier is imposed due to the organic-to-inorganic carrier transition mechanism, as illustrated in Figure R4c. These energy landscapes agree with the I-V symmetry characteristic analysis in Figure R2.

Accordingly, we have made revisions in the manuscript to strengthen the argument. The revision is on page 11 in the revised manuscript, as below:

“In the dark, the 337 nm-thick BA₂PbI₄ film exhibits a quasi-exponential ($R^2= 0.956$) decrease in the ($I_{\text{for}}+I_{\text{rev}}$) as the applied bias V_{abs} increases (Fig. 2B). The imperfect exponential dependence indicates the presence of weak built-in potentials at contacts, which can be attributed to the slight difference in built-in potential between the near-ITO side and the near-Cu side, as illustrated in fig. S14. The built-in potential disparity is fitted to be 57.8 ± 7.2 meV, which aligns with the reasonable work function difference between the Cu and ITO electrodes. Intriguingly, this I-V dependence reverses upon simulated solar illumination. The ($I_{\text{for}}+I_{\text{rev}}$) instead increases exponentially with V_{abs} , and it follows a perfect exponential dependence ($R^2= 0.993$), suggesting the emergence of an inverse ΔV upon illumination, which is consistent with the effect of the interfacial dipole.”

We sincerely hope the above reply addresses your concern.

Comment #3. While the conclusion briefly summarizes the UV-induced de-passivation mechanism of ammonium ligands and mentions its potential implications for other phenomena (e.g., ion migration, phase separation), these discussions lack in-depth exploration and data support in the main text, appearing vague.

Response: Thank you for your helpful comment. We do believe the light-induced de-passivation mechanism in this work can contribute to the understanding of many important phenomena in this field, including the dynamism of the 2D/3D perovskite interface, phase segregation in mixed-halide perovskites, ion migration and hysteresis in devices, enhancements in photocatalytic activity, and the photoluminescence blinking/quenching of perovskite nanocrystals. While we acknowledge this claim is largely based on our understanding of perovskites rather than data, we would also like to clarify that the theme of this work is to introduce this unknown mechanism. The statement you commented on is for discussion, not an argument, which is why we place it in the Discussion section. Proving these phenomena requires huge efforts, and we believe it is unrealistic to include them in this self-contained work.

To make a clearer statement, we have expounded this discussion in the revised manuscript (on pages 21-22). The text is shown below:

“We believe the mechanism revealed here could also contribute to the understanding of many other important phenomena in this field, such as the dynamism of the 2D/3D HP interface,^[13, 53, 54] phase segregation in mixed-halide HPs,^[4] ion migration and hysteresis in devices,^[55] enhancements in photocatalytic activity,^[43] and the photoluminescence blinking/quenching of HP nanocrystals.^[56] Specifically, the 2D/3D HP configuration, wherein a 2D HP layer serves as a protective and passivating coating over a 3D HP absorber, exhibits a dynamic dimensionality change at the interface over time.^[54] This dynamism causes deviations from the initial design and compromises device efficiency and stability. This dynamic behavior leads to deviations from the initial structural design, compromising both device efficiency and stability. Previous studies have observed the migration of ammonium ligands under illumination, consistent with the mechanism proposed herein;^[57] Ion migration, a prevalent phenomenon in HPs, underpins critical device behaviors such as hysteresis.^[55] While

anchoring ammonium ligands can mitigate ion migration, this effect may be exacerbated under illumination.^[58] This photoresponse aligns with the proposed mechanism; For photocatalytic applications of 2D HPs, such as H₂ evolution,^[43] efficient separation of electrons and holes is essential. Our mechanism suggests that this separation is facilitated by an organic-to-inorganic carrier transition induced by violet/UV photons, thereby enhancing photocatalytic activity; Additionally, ammonium ligands are commonly employed as surface-passivating agents for HP nanocrystals. However, a light-induced depassivation mechanism may trigger ligand dynamics at the surface, exposing active sites and introducing trap states. This process could contribute to charge trapping/detrapping phenomena, resulting in photoluminescence blinking or quenching.^[56] Nevertheless, further experimental investigations are required to substantiate these hypotheses.”

References

4. Meggiolaro, D., Mosconi, E. & De Angelis, F. Formation of surface defects dominates ion migration in lead-halide perovskites. *ACS Energy Lett.* **4**, 779-785 (2019).
13. Park, S. M. et al. Engineering ligand reactivity enables high-temperature operation of stable perovskite solar cells. *Science* **381**, 209-215 (2023).
43. Wang, H. et al. Mechanistic understanding of efficient photocatalytic H₂ evolution on two-dimensional layered lead iodide hybrid perovskites. *Angew. Chem., Int. Ed.* **60**, 7376-7381 (2021).
53. Teale, S., Degani, M., Chen, B., Sargent, E. H. & Grancini, G. Molecular cation and low-dimensional perovskite surface passivation in perovskite solar cells. *Nat. Energy* **9**, 779-792 (2024).
54. Sutanto, A. A. et al. Dynamical evolution of the 2D/3D interface: a hidden driver behind perovskite solar cell instability. *J. Mater. Chem. A* **8**, 2343-2348 (2020).
55. Alharbi, E. A. et al. Atomic-level passivation mechanism of ammonium salts enabling highly efficient perovskite solar cells. *Nat. Commun.* **10**, 3008 (2019).
56. Tian, Y. et al. Giant photoluminescence blinking of perovskite nanocrystals reveals single-trap control of luminescence. *Nano Lett.* **15**, 1603-8 (2015).
57. Mathew, P., Cho, J. & Kamat, P. V. Ramifications of ion migration in 2D lead halide perovskites. *ACS Energy Lett.* **9**, 1103-1114 (2024).
58. deQuilettes, D. W. et al. Photo-induced halide redistribution in organic-inorganic perovskite films. *Nat. Commun.* **7**, 11683 (2016).

We believe the mechanism reported in our work will provide a new perspective for the above phenomena and contribute to advancing their understanding.

We sincerely hope the above reply addresses your concern.

Comment #4. The photocatalytic hydrogen production experiment (Figure 3) indirectly speculates deprotonation of ammonium ligands through H₂ monitoring. Why weren't direct characterizations (e.g., XPS or NMR) to confirm the chemical conversion from -NH₃⁺ to -NH₂?

Response: Thank you for your insightful comment. We tried to organize our manuscript in a step-by-step logic. First, we revealed the **HOMO-to-E_x (organic-to-inorganic) carrier transition (Figure 1-2 in the manuscript)** under violet/ultraviolet excitation with the transient absorption spectrum characterization and the steady-state optoelectronic characterization; Then, we designed the photoreaction system to prove the ensuing **deprotonation process (Figure 3 in the manuscript)**; Finally, we confirmed the **disruption of the ligand-perovskite interaction and its impact on solar cells (Figure 4 in the manuscript)** that causes depassivation. In this way, we hope to present this mechanism clearly. The Figure 3 you mentioned is used to confirm the deprotonation process, while the change of the -NH₃⁺ is related to the disruption of the ligand-perovskite interaction.

To directly validate the disrupted ammonium ligand-perovskite interaction, we have carried out experiments on both 2D and 3D perovskite systems. The details of the 3D perovskite system by the Fourier Transform Infrared (FTIR) characterization are introduced in the response to Comment #6 below. The details of the 2D perovskite system are as follows.

Figure R5. Disrupted ammonium-perovskite interaction after (near) UV light-driven deprotonation. (a) Schematic diagram of electron transitions upon the ~2.76 eV (450 nm) and ~3.40 eV (365 nm) monochromatic excitations. Comparative absorption spectra of the BA₂PbI₄ films at 0 min and 60 min upon (b) the ~2.76 eV irradiation and (c) the ~3.40 eV irradiation, respectively. The intensities of the two irradiations are both 20 mW cm⁻². The test was implemented in the ambient air. (d) Comparison of the Urbach energy evolutions of the 2D perovskite film in different illumination conditions (~3.40 eV, ~2.76 eV, and darkness). Overall, the evolving absorption profile and the increase in Urbach energy agree with the (near) UV light-induced deprotonation and detachment of the ammonium from the perovskite framework.

We carried out a comparative irradiation test on BA₂PbI₄ perovskite films and used the X-ray diffraction (XRD) measurement to characterize the products. Specifically, we irradiated BA₂PbI₄ films with monochromatic lights of ~2.76 eV (450 nm) and ~3.40 eV (365 nm), respectively. The ~2.76 eV irradiation is for the band-edge photoexcitation, while the ~3.40 eV irradiation provokes the ammonium-to-perovskite carrier transition (Figure R5a). The irradiation intensities of both sources were

controlled to be 20 mW cm^{-2} . As shown in Figure R5b, the absorption profile of the film by the $\sim 2.76 \text{ eV}$ irradiation barely changed within the first 60 minutes, indicating high photostability. In stark contrast, the $\sim 3.40 \text{ eV}$ irradiation caused a conspicuous decline in the resonance intensity (Figure R5c) after 60 min. Meanwhile, the absorption edge also became blunted after the irradiation, implying the appearance of mid-gap trap states.

Taking the Urbach energy as the quantitative metric, the $\sim 2.76 \text{ eV}$ and $\sim 3.40 \text{ eV}$ irradiations impose vastly different degradation kinetics on the BA_2PbI_4 films. As compared in Figure R5d, the degradation upon the $\sim 3.40 \text{ eV}$ irradiation started immediately and featured a short τ of 12.2 min. However, a discernible degradation upon the $\sim 2.76 \text{ eV}$ irradiation only appeared after 60 min and featured a much longer τ of 55.6 min. **Such a difference in the characteristic time indicates distinct decomposition routes triggered by the two irradiations.** The mild photolysis by the band-edge photoexcitation was unveiled previously to arise from a photooxidation process, while **the acute photodegradation by the $\sim 3.40 \text{ eV}$ irradiation could be assigned to the (near) UV light-activated deprotonation and detachment of the ammonium.**

Fig. R6. The XRD pattern of the BA_2PbI_4 film after the $\sim 3.40 \text{ eV}$ (365 nm) irradiation for 60 min.

The XRD characterization was then conducted to analyze the products of the BA_2PbI_4 film exposed to the (near) UV irradiation (Figure R6). BAI was detected, agreeing with the disrupted BA-perovskite interaction arising from our mechanism.

To summarize, the photoreaction system, designed with DFT simulations, is used to prove the deprotonation caused by our mechanism, contributing to a step-by-step argument. The direct characterizations (XRD, FTIR) are also provided, but in the following sections. Thank you very much for your understanding.

We sincerely hope the above reply addresses your concern.

Comment #5. What scientific question does the absorption spectrum data in Figure 4B aim to address? Please elaborate on this section.

Response: Thank you for your helpful comment. Figure 4B (shown below as Figure R7a) is the steady-state UV-vis absorption spectrum of a MAPbI_3 perovskite film surface-treated with BAI. **It shows that after the surface treatment, an enhanced absorption can be found in the range of 300 to 400 nm (corresponding to photon energies between 3.10 eV and 4.13 eV), which aligns with our proposed ligands-to-perovskite carrier transition mechanism.** The higher the concentration of ammonium, the stronger the characteristic absorption, as can be checked in Fig. S32a in the Supplementary Information file, and is present as Figure R7b below.

Figure R7. (a) Comparative absorption spectra of 3D perovskite (MAPbI_3) films before

and after surface treatment with ammonium ligands (butylamine iodide). (b) Steady-state absorption spectra of 3D perovskite (MAPbI₃) films surface-treated by butylamine iodide solutions with a gradient of concentrations.

The UV-vis absorption characterization is to confirm the viability of the mechanism in the ligand-passivated 3D perovskites. Prior characterizations in the manuscript are carried out in (quasi-) 2D perovskites considering the abundant ligand-perovskite bonds and the easy characterizations, and it is important to extend this mechanism to the 3D perovskite system.

We have provided the corresponding statement in the prior manuscript version for clarification (on page 18 in the manuscript), as shown below:

“To support this point, we compared the absorption spectra of a pristine 3D HP (MAPbI₃) film and a surface-treated 3D HP film with BA in Fig. 4B. To avoid interference from 2D HPs, the dose of ammonium-terminated ligands (BAI in this case) for surface treatment was carefully controlled. The absence of 2D HPs is demonstrated in fig. S32. The two absorption spectra exhibit perfect overlap within the tested wavelength range, except between 300 and 400 nm (corresponding to photon energies between 3.10 eV and 4.13 eV), where the absorption of the ligand-treated film is significantly enhanced compared to the pristine film. Since MAPbI₃ is mono-cationic (containing only the CH₃NH₃⁺ cation, no Cs⁺ or CH(NH₂)₂⁺) and mono-anionic (containing only the I⁻ anion), the observed enhanced absorption is unlikely to arise from phase separation. Moreover, the enhanced absorption does not originate from BAI (as evidenced in fig. S3). Instead, the energy range of the enhanced absorption coincides with the violet/UV light-induced ammonium-to-HP electron transition mechanism. Furthermore, consistent with the universality of the mechanism, the enhanced absorption in the 300 to 400 nm range is also observed in 3D HP films treated with other types of ammonium-terminated ligands, as shown in fig. S33. These results confirm the applicability of the proposed mechanism to 3D HPs. Notably, resolving the ammonium-to-HP electron transition signal in surface-treated 3D HP films using TA characterization is challenging due to its overlap with the intrinsic high-energy absorption signals of 3D HPs, as well as difficulties in eliminating interference from 2D HPs (fig. S34).”

We sincerely hope the above reply addresses your concern.

Comment #6. In Figure 4C, the weakened ligand signals in FTIR alone are insufficient to convincingly prove the disruption of ammonium-HP interactions and attribute this process to ligand deprotonation. Please provide further justification.

Response: Thank you for your insightful comment. To verify the disrupted ligand-perovskite lattice interaction, we have carried out characterizations in both 2D and 3D perovskite systems. For the 2D perovskites, we have carried out irradiation tests and XRD characterizations, as detailed in the response to Comment #4. The FTIR characterization you commented on is used to confirm the disrupted ligand-perovskite interaction in the 3D perovskite system.

While we totally agree that the prior FTIR characterization result is not convincing enough, we do believe it is direct evidence that shows the detachment of ligands from the perovskite surface due to the photoinduced depassivation mechanism. To further strengthen this argument, we have measured the irradiation time-dependent FTIR spectrum. The dynamic FTIR characterization should be more convincing than the prior static FTIR characterization. The details are introduced below.

MAPbI₃ was selected as the perovskite sample as it is single-anion, single-cation, and phase-stable at room temperature. Otherwise, there could be a phase segregation or a phase transition that interferes with the result interpretation. The ligand pair of butylammonium and oleic acid was used as the passivating agent. For the experiment, the MAPbI₃ polycrystalline powder was dispersed in a benign chlorobenzene solution containing the ligands. The doses are as follows: 100 mg MAPbI₃ polycrystalline powder was dispersed in 5 mL chlorobenzene that contains 11.9 mg butylammonium and 45.8 mg oleic acid. The molar ratio of butylammonium to oleic acid is 1:1. The solution was gently stirred by a magnetic stirrer during the test. Meanwhile, a monochromatic illumination of 365 nm was applied to the solution with a power density of ~25 mW/cm². The 365 nm irradiation is sufficient to trigger the depassivation mechanism. The dispersion was collected every hour. Then, the MAPbI₃ powders were isolated by centrifugation and vacuum drying. In comparison, in another set, the same MAPbI₃ dispersion was subjected to the same test except that it was covered with

aluminum foil to avoid irradiation. The irradiated sample with different irradiation times, the non-irradiated sample, and the pristine MAPbI₃ sample were sent for the FTIR measurement.

Figure R7. The irradiation time-dependent FTIR result of the passivated perovskite sample.

As shown in Figure R7, the FTIR characterization confirms the gradual detachment of ligands under the 365 nm irradiation. The 2854 cm⁻¹ peak arises from the symmetric stretching vibration of -CH₂- groups, which originate from both the butylammonium and oleic acid segments in the ligand pair, while it is not detected in MAPbI₃. Therefore, this characteristic FTIR peak can be used to monitor the residue of ligands on the surface of MAPbI₃. It should be noted that the peak of the N-H scaling vibration may not be an ideal observation target, as the MAPbI₃ perovskite intrinsically shows a strong N-H scaling vibration signal as well.

Importantly, as the irradiation time increases, this characterization peak of -CH₂- wanes gradually, indicating the detachment of ligands from the perovskite surface. This is in line with the photoinduced depassivation mechanism introduced in our work. After 5 h of irradiation, this characteristic peak becomes barely discernible. In contrast, for the

non-irradiated sample, the 2854 cm^{-1} peak remains after the 5h test. This contrast rules out the impact of physical detachment and confirms that the 365 nm irradiation is the driving factor of the ligand detachment, agreeing with the photoinduced detachment mechanism.

We would also like to clarify that it is challenging to characterize the ligand detachment in the 3D perovskite system. This is because the dose of ligands needs to be prudently controlled. Otherwise, 2D perovskite would form to interfere with the analysis. For example, in the UV-vis characterization discussed in the response to Comment #5, the doses of ammonium ligands for the surface treatment were controlled to prevent the formation of 2D perovskites, so the enhanced absorption in the UV range can be ascribed to the ligand-to-perovskite carrier transition. The low content of ligands on the surface of 3D perovskites makes them difficult to detect. To support this claim, we have carried out X-ray Photoelectron Spectroscopy (XPS) characterizations on the surface passivated MAPbI₃ films, as shown in Figure R8 below.

Figure R8. XPS characterization of surface-treated MAPbI₃ films with butylammonium iodide and pristine MAPbI₃ films for (a) the Pb element, (b) the I element, and (c) the N element.

The concentration of the butylammonium for the surface treatment is 2.5 mg/ml (dissolved in isopropanol), which is close to the upper limit for avoiding the formation of 2D perovskites based on our observation. However, the XPS spectra show negligible difference between the surface-treated and pristine MAPbI₃ films. This result shows it can be challenging to detect the ammonium ligands even for the XPS characterization.

Therefore, the dynamic FTIR characterization in the 3D perovskite system, along with the XRD characterization in the 2D perovskite system, serves to confirm the disrupted interaction between ammonium ligands and the perovskite lattice due to the

depassivation mechanism. Accordingly, we have made revisions in the revised manuscript (on pages 18-19), as shown below:

“Consequently, exposure to violet/UV photons causes the deprotonation of ammonium ligands, disrupting the ammonium-HP interaction. To verify this, we conducted Fourier-transform infrared (FTIR) spectroscopy on both UV-irradiated (365 nm @ 25 mW/cm² for 5 h) and non-irradiated ammonium-passivated MAPbI₃ samples. During irradiation, the 3D HP powder was gently stirred and dispersed in chlorobenzene with ligands. It is well established that HPs adopt an X-type ligand binding configuration, where Lewis acid and Lewis base moieties attach to the surface of HP in pairs.^[32, 46] Herein, we employed BA and oleic acid (1:1 in molar ratio) as the ligand pair. The 3D HP powder was collected every one hour, followed by isolation for the FTIR characterization. As shown in Fig. 4C, the 2854 cm⁻¹ peak arises from the symmetric stretching vibration of -CH₂- groups. It originates from both the butylammonium and oleic acid segments in the ligand pair but is not detected in MAPbI₃. Notably, as the irradiation time increases, the -CH₂- peak wanes gradually, indicating the detachment of ligands from the HP surface. After 5 h of irradiation, this characteristic peak becomes barely discernible. In contrast, for the non-irradiated sample, the 2854 cm⁻¹ peak remains after the 5h test. This contrast confirms that the 365 nm irradiation is the driving factor for ligand detachment, in agreement with our mechanism.”

In the end, we sincerely thank you again for dedicating your valuable time to reviewing our work and for providing insightful feedback. We hope our responses adequately address your concerns and align our manuscript with the high standards of *Nature Communications*. We have made every effort to refine our work and welcome any further comments or suggestions in subsequent reviews. Thank you once again for your thoughtful contributions.

Response to Reviewer #2:

Research by Yu et al. discovered that violet/ultraviolet light can induce depassivation of ammonium -terminated ligands. These ligands become deprotonated, thereby leading to the depassivation phenomenon. This provides a novel insight into the violet/ultraviolet light-induced degradation of highly efficient perovskite solar cells prepared with meticulous passivation, aiding in understanding the dynamic behavior at the two-dimensional/three-dimensional efficient photovoltaic interface. However, the experimental evidence for some key conclusions still needs to be strengthened, especially in terms of the universality of the mechanism, the characterization of dynamic processes and the influence of environmental factors. While the perspective offered by this research work is novel, several points require clarification, as detailed below:

Comment #1. The authors claim that violet/ultraviolet light induces the deprotonation of amines, and this deprotonation disrupts the ligand-perovskite interaction, triggering the depassivation of the perovskite making it susceptible to light degradation. However, the issue is that UV light itself can induce the degradation of perovskite, regardless of whether it is passivated or not. Furthermore, UV light also damages electron/hole transport layer materials. The reduction in perovskite solar cell performance is the result of the combined effects of UV light.

Response: We sincerely thank you for dedicating your valuable time to reviewing our manuscript and for providing insightful comments. We have carefully revised the manuscript to address your feedback to the best of our ability. We hope the refined version more effectively conveys our scientific findings and contributes meaningfully to the perovskite research community.

We totally agree that the harm caused by UV lights to perovskite solar cells is not limited to the depassivation reported here, but also to the perovskite itself (passivated or not) and charge-transport materials. Therefore, we organize our manuscript in a step-by-step logic. **First**, we revealed the carrier dynamics to reveal the ammonium-to-perovskite carrier transition (Figure 1 and Figure 2 in the manuscript). **Second**, we confirmed the cause of deprotonation of the ammonium group (Figure 3). **Finally**, we showed disrupted ammonium-perovskite interaction (depassivation) and its impact on

perovskite solar cells (Figure 4). By this step-by-step argument, we focus on the violet/ultraviolet light-induced depassivation mechanism, minimizing the interference from other parallel mechanisms.

Furthermore, when it comes to the solar cell device performance characterization, which is to help validate the depassivation mechanism, the impact of parallel mechanisms caused by UV irradiation should be minor in our work. The reasons are detailed below:

First, as you commented, UV lights also jeopardize charge-transport layers and the perovskite itself. In this work, since a p-i-n inverted device configuration is used, the impact on the hole-transport layer (SAM, 4PACZ) should be considered. According to previous reports, the jeopardization arises from 1) the decomposition of aromatic rings in SAM molecules and the debonding between the SAM and ITO (*Sci. Adv.* **11**, eadu3493 (2025)), and 2) the degradation of the SAM/perovskite interface (*Science* **384**, 1126-1134 (2024)). **However, these harms are reported to only become discernible in hours or tens of hours.** In our test protocol with intermittent light-dark alternation, the solar cell in every cycle was first illuminated for an I-V scan that lasts 7 seconds and was then allowed to recover in the dark for 3 seconds for self-healing. The number of cycles implemented was only several hundred cycles, amounting to an irradiation time of **less than one hour.** **Therefore**, during our test, the parallel impacts of UV lights should be minor.

Second, we have carefully designed the solar cell performance testing protocol, *i.e.*, the light-dark alternation testing protocol, to minimize the impact from parallel mechanisms induced by UV irradiation. The details are as introduced in Supplementary Text 5 in the Supplementary file, as shown below:

“Supplementary Text 5:

It should be noted that UV lights affect HPs in several ways. Previous reports showed that long-term, continuous UV irradiation would result in irreversible chemical decomposition of HPs.^[9, 10] Short-term illumination would generate metastable mid-gap states in HPs, but these metastable states could disappear after a self-healing process in darkness.^[11, 12] To highlight the mechanism proposed here, the testing method needs to minimize interference by other parallel impact mechanisms. Therefore, we adopted the short-term, alternating illumination/darkness test method proposed by Motti et al with modifications.^[13]

In detail, the solar cell was subjected to a short-term (7 s) illumination followed by a recovery process in darkness (3 s). During illumination, a defect annihilation process and a defect generation process would be triggered simultaneously:

- 1) The photo-driven annihilation of Frenkel defects ($I_i^- - V_I^+$ Frenkel pairs) that enhances the photovoltaic performance.^[13, 14]
- 2) The reaction between two filled traps I_i^0 to form one I_2 molecule that compromises the photovoltaic performance.^[13]

The defect generation process tends to occur at a defective region where the I^- is poorly stabilized, but can be suppressed upon passivation by ammonium-terminated ligands. The two processes are ion-migration-related, thus are cumulative and barely reversible in darkness.^[13] Therefore, the alternating testing can minimize the interference from parallel mechanisms.

UV lights could also jeopardize SAMs, decomposing their aromatic rings and weakening the interaction between them and ITO.^[15] However, the harm to SAMs by UV irradiations only becomes discernible in hours or tens of hours,^[15, 16] while in our test protocol with intermittent light-dark alternation, the solar cell in every cycle was first illuminated for an I-V scan that lasts 7 seconds, and was then allowed to recover in the dark for 3 seconds for self-healing. A significant distinction in performance degradation was observed after 47 cycles (as shown in Figure 4E in the main text), amounting to only 329 s of irradiation. During this time slot, the impact of UV lights on the SAM layer could be minimal.

Based on the above discussion, the testing protocol here allows effective isolation of the light-induced depassivation effect on HP solar cells.”

References:

9. Farooq, A. et al. Spectral dependence of degradation under ultraviolet light in perovskite solar cells. *ACS Appl. Mater. Interfaces* **10**, 21985-21990 (2018).
10. Wang, Y. et al. Tautomeric molecule acts as a "sunscreens" for metal halide perovskite solar cells. *Angew Chem. Int. Ed. Engl.* **60**, 8673-8677 (2021).
11. Nie, W. et al. Light-activated photocurrent degradation and self-healing in perovskite solar cells. *Nat. Commun.* **7**, 11574 (2016).

12. Khenkin, M. V. et al. Dynamics of photoinduced degradation of perovskite photovoltaics: from reversible to irreversible processes. *ACS Appl. Energy Mater.* **1**, 799-806 (2018).
13. Motti, S. G. et al. Controlling competing photochemical reactions stabilizes perovskite solar cells. *Nat. Photonics* **13**, 532-539 (2019).
14. Mosconi, E., Meggiolaro, D., Snaith, H. J., Stranks, S. D. & De Angelis, F. Light-induced annihilation of Frenkel defects in organo-lead halide perovskites. *Energy Environ. Sci.* **9**, 3180-3187 (2016).
15. Liu, T. et al. Efficient perovskite solar modules enabled by a UV-stable and high-conductivity hole transport material. *Sci. Adv.* **11**, eadu3493 (2025).
16. Fei, C. et al. Strong-bonding hole-transport layers reduce ultraviolet degradation of perovskite solar cells. *Science* **384**, 1126-1134 (2024).

We sincerely hope the above reply addresses your concern.

Comment #2. When measuring the device IV curves under filtered/non-filtered light conditions, the performance decline in non-filtered devices is primarily due to reductions in V_{oc} and FF, while J_{sc} is less affected. Why is this? Correspondingly, the authors show in Figure 4D that the short-circuit current density of the unpassivated device is only 22.09 mA/cm², while that of the passivated device significantly increases to 26.22 mA/cm². How do the authors explain this?

Response: Thank you for your insightful comment.

FIRST, your comment pointed out a careless statement we made in the main text, as detailed below.

Figure R9 (Fig. S38 in the manuscript). Comparative residual I-V characteristics of the HP solar cell after being continuously illuminated by filtered and unfiltered lights for 24 h.

The J-V characteristics shown in Figure R9 (Fig. S38 in the manuscript) are those of the devices **aged** under filtered and unfiltered illuminations. They show larger distinctions in V_{oc} and FF, while J_{sc} values are less discrepant. However, compared to fresh devices, **all J_{sc} , V_{oc} , and FF have significantly declined after aging**, as shown in Figure R10 below.

Figure R10. I-V characteristic evolutions of the perovskite solar cells subjected to (a) filtered illumination and (b) unfiltered illumination. For filtering the illumination, a 420 nm long pass is used.

Therefore, it is incorrect for us to claim that “ J_{sc} is less affected” in the main text. The correct statement should be “ J_{sc} is less discrepant compared to V_{oc} and FF after the aging”. We made a careless, misleading statement in the main text. But this was not intentional, as we stated it correctly in Supplementary Text 6:

“After 24 hours, the solar cell under unfiltered illumination exhibits a significant decline in V_{oc} to 0.906 V and FF to 57.15%, while the device under filtered illumination retains a much higher V_{oc} of 1.103 V and an FF of 64.72%. The residual J_{sc} values are **less distinct** between the two sets of solar cells, measuring 19.23 mA/cm² for the unfiltered and 20.50 mA/cm² for the filtered illumination.”

We are so sorry for this mistake and have revised the statement in the new manuscript version, as shown below (on page 21):

“In another test protocol, after being subjected to continuous illumination for 24 h, the unfiltered device exhibits a more severe decline in V_{oc} and FF compared to the filtered device (fig. S38 and Table S1), a pattern indicating defect formation as V_{oc} and FF are sensitive to defects (Supplementary Text 6).”

In principle, J_{sc} is less sensitive to defects compared to V_{oc} and FF. This is because J_{sc} is largely determined by light absorption and charge transport, where defects primarily

cause recombination with minimal impact on these processes. In contrast, V_{oc} and FF are highly sensitive to defects due to their strong dependence on recombination losses. The reason the aging causes the J_{sc} decline here can be ascribed to defect-induced degradation of the interface between the perovskite layer and the charge transport layers (*ACS Nano* **10**, 218-224 (2016)).

SECOND, the reason the unpassivated device shows a much lower J_{sc} than the passivated device is that the ammonium ligands, besides serving as passivating agents, also play a role in the crystallization process and improve the crystallinity of the perovskite layer (*Nature* **616**, 724-730 (2023)).

In our recipe, as we stated in the experimental section, we used octane-1,8-diammonium and phenylpropylammonium as the passivating agents. Octane-1,8-diammonium is incorporated in the perovskite precursor for bulk passivation, and phenylpropylammonium is for surface passivation. Besides bulk passivation, octane-1,8-diammonium also modulates the crystallization of the perovskite layer to improve its crystallinity (*Adv. Mater.* **36**, e2400105 (2024)). Assuming smooth carrier extraction, J_{sc} of the solar cell is decisively determined by the light absorption and carrier transport of the perovskite layer. A higher crystallinity of the perovskite layer would contribute to both light absorption and carrier transport. Therefore, the passivated device shows a much higher J_{sc} than the unpassivated device.

To clarify, we have added the following statement in the revised manuscript (on pages 19-20), as shown below:

“As shown in Fig. 4D, the pristine solar cell without any ammonium-based passivation exhibited a champion PCE of 17.23% (open-circuit voltage $V_{oc} = 1.10$ V, short-circuit current $J_{sc} = 22.09$ mA/cm², fill factor FF = 70.71%). In contrast, proper passivation of the HP layer using PPA and octane-1,8-diammonium significantly improved the device performance, resulting in a champion PCE of 26.44% ($V_{oc} = 1.179$ V, $J_{sc} = 26.22$ mA/cm², FF = 85.49%) in our lab. The certified PCE by a third party is 26.15% (for reverse scan, $V_{oc} = 1.188$ V, $J_{sc} = 25.86$ mA/cm², FF = 85.12%, fig. S35). This contrast confirms the beneficial passivation effect induced by the ammonium ligands. **Part of the improvement also arises from the crystallization modulation effect of the ammonium ligands.**^[50]”

References

50. Jia, P. et al. Intermediate phase suppression with long chain diammonium alkane for high performance wide-bandgap and tandem perovskite solar cells. *Adv. Mater.* **36**, e2400105 (2024).

We sincerely hope the above reply addresses your concern.

Comment #3. The authors propose the phenomenon of violet/ultraviolet light-induced depassivation in passivated perovskite solar cells but do not provide specific methods to address this problem in order to obtain UV-stable perovskite solar cells.

Response: Thank you for your insightful comment.

This work focuses on unveiling the vulnerability of the widely used ammonium ligands against violet/ultraviolet irradiation and the underlying mechanism. Although we are unable to offer immediate solutions, this work could open the avenue to designing the next-generation passivating agents to advance the operational durability of perovskite solar cells.

By taking a close look at the depassivation mechanism reported here, it works because violet and ultraviolet photons are sufficient to pump the carrier transition from the ammonium HOMO to the perovskite E_x /CBM states. Therefore, the light-induced depassivation can be prevented if the HOMO- E_x /CBM gap increases beyond the violet-ultraviolet photo energy. Also, given that this mechanism renders a trend for depassivation, multi-site anchoring could also alleviate the light-induced depassivation. Accordingly, we believe the following directions might merit consideration:

- 1) Ammonium ligands exhibit adjustable chemical properties. Functional groups on these ligands can be engineered to increase the ionization potential of the ammonium moiety.

- 2) Given that this mechanism tends to disrupt the interaction between ammonium ligands and the perovskite lattice, multi-site anchoring could help alleviate the light-induced depassivation.
- 3) Instead of using ammonium ligands directly as passivating agents, 2D perovskites can be a sturdier alternative. 2D perovskites can be tailored not only in the ammonium ligands but also in the perovskite sublattice. By decreasing the electron affinity of the perovskite lattice (e.g., by using non-lead 2D perovskites or alloying), the HOMO- E_x /CBM gap can also be increased.
- 4) New types of ligands beyond ammonium-terminated ligands should be developed for perovskite solar cells. For instance, phosphorus-, oxygen-, and sulfur-centered ligands, such as phosphonium salts, carboxylic acids, and thiols, show great promise. They are not as widely applicable in perovskites as ammonium ligands, warranting further exploration.

We have added the above discussion in the revised manuscript in the “**Discussion**” section (on pages 23-24). We hope this modification addresses your concern and helps inspire possible solutions. Thank you for your understanding.

We sincerely hope the above reply addresses your concern.

Comment #4. It would be beneficial if the authors could provide the EQE of the perovskite solar cells before and after UV aging, along with corresponding analysis, to gain a deeper understanding of the perovskite's photo-response.

Response: Thank you for your insightful comment. We have reimplemented the ageing test to supplement data in response to this comment. Below, we show the EQE data for the UV aging in both the ambient air and the N₂-filled glovebox. Other complementary data can be found in responses to other comments.

For the ageing test carried out in the ambient air, the depassivation mechanism has a synergetic effect with the air. This is because the depassivation mechanism pumps the electrons from the HOMO level in the ammonium ligands to the CBM/ E_x states in the perovskite lattice, forming an excited state that is vulnerable to attack by redox

substances in the air, such as oxygen.

Figure R11. The I-V characteristics of passivated solar cell devices subjected to (a) the 365 nm irradiation and (b) the 450 nm irradiation in the ambient air, respectively. (c)-(d) The corresponding EQE spectra for the two devices before and after the ageing test.

We carried out the tests in a fume hood with two irradiation sources (365 nm and 450 nm) for comparison. The 365 nm irradiation can trigger the depassivation mechanism, while the 450 nm irradiation cannot. The powers of both irradiations were set to 20 mW/cm². We irradiated both devices for 30 min and then restored them in a N₂-filled glovebox in the dark for 5 h.

As shown in Figure R11a and Figure R11b, the J_{sc} of both devices barely changes. The impact of the irradiations is reflected in V_{oc} and FF. Notably, the FF of the device subjected to the 365 nm irradiation shows a much more serious decline in FF (from 85.16% to 69.98%) compared to the device subjected to the 450 nm irradiation (from 85.19% to 76.60%). This observation suggests the generation of more defects by the 365 nm irradiation, which is in line with the depassivation mechanism.

According to the EQE spectra in Figure R11c and Figure R11d, although the J_{sc} data do not exhibit a conspicuous change for both devices, the EQE spectrum of the 365 nm-irradiated device shows that the EQE values decrease in the short-wavelength region but increase in the long-wavelength region. This could be ascribed to slight phase segregation that changes the light-responsive pattern. As such, this phenomenon agrees with the depassivation mechanism, as the passivation failure will destabilize ions to facilitate phase segregation.

Figure R12. EQE spectra of solar cell devices subjected to (a) a simulated AM 1.5G illumination with photons of <420 nm wavelength filtered and (b) a non-filtered, full-spectrum AM 1.5G irradiation.

For tests carried out in an N_2 -filled glovebox, the solar cell devices were subjected to two illumination conditions for comparison: 1) simulated AM 1.5G illumination with photons of <420 nm wavelength filtered; 2) non-filtered, full-spectrum AM 1.5G illumination. The EQE spectra of the devices before and after the ageing test are shown in Figure R12. Consistent with the results obtained in the ambient ageing test, the J_{sc} barely changes after the ageing test in the N_2 -filled glovebox. Besides, due to the absence of air, the phase segregation phenomenon is also alleviated, and the EQE profiles for both devices barely change before and after the test.

The above EQE data reveal that J_{sc} is less sensitive to the depassivation mechanism, agreeing with the statement in our manuscript. This is because J_{sc} is essentially less sensitive to defects compared to V_{oc} and FF. The barely impacted J_{sc} also confirms that the tests we carried out do not cause a comprehensive degradation of the perovskite solar cell, making the argument of defect depassivation more convincing.

We sincerely hope the above reply addresses your concern.

Comment #5. From the data in Figure 4E, it can be seen that under illumination of 1.064 suns, the performance of the perovskite solar cell decreased by nearly 10% in less than 300 seconds of light exposure. This cannot simply be attributed to ultraviolet light in the solar spectrum; the perovskite cell itself has severe intrinsic light stability issues. Even under UV-filtered light, the performance of the perovskite cell also showed a significant decrease within the same timeframe, dropping from a peak of about 1.05 to around 0.98.

Response: Thank you for your insightful comment, which helps us improve our argument.

To understand this phenomenon, the mechanisms underlying the testing protocol warrant clarification, which is detailed in the response to Comment #1. To isolate the effect of the depassivation, we adopted an intermittent testing protocol with light-dark alternations. The solar cell in every cycle was first illuminated for 7 seconds, **during which an I-V scan was applied**, and was then allowed to recover in the dark for 3 seconds for self-healing. As detailed in the response to comment #1, a defect annihilation process (annihilation of $I_i^-V_i^+$ Frenkel pairs) and a defect generation process (two I_i^0 combining into I_2) would be triggered simultaneously during the illumination period, and both are ion migration-related (*Nat. Photonics* **13**, 532-539 (2019)).

The defect generation process tends to occur at a defective region where iodide ions are poorly stabilized (*Nat. Photonics* **13**, 532-539 (2019)). According to the depassivation mechanism reported here, violet/ultraviolet lights will disrupt the ligand-perovskite interaction, so the depassivated sites (including I sites) would lose anchoring and tend to migrate. **The I-V scans are applied not only to record the performance evolution but, more importantly, also to drive the ion migration**, so that the defect evolution can be reflected in performance.

Figure R13. Schematic diagram of the dynamic energy band bending for a heterojunction during I-V scans.

Notably, the cyclic I-V scan test is not like the maximum power point (MPP) tracing test commonly adopted in perovskite solar cells. It is a dynamic testing condition. As illustrated in Figure R13, when at V_{oc} , the built-in potentials are mitigated by the applied bias voltage to the extent that the drift current equals the diffusion current; when at J_{sc} , no bias voltage is applied, and the built-in electric fields are at their maxima. Therefore, during the I-V scans, ions within the depletion region would be subject to dynamic built-in electric fields. As such, ion migration is more easily triggered compared to that in an MPP tracing test, in which the built-in electric field is constant.

We have made modifications in the manuscript on page 20 to clarify, as shown below (fig. S36):

“The built-in potentials of the heterojunctions vary during the I-V scan in each replication, causing periodic changes in the built-in electric field that affect the ions in the depletion region (fig. S36).”

Additionally, we would also like to clarify that the common MPP test was not adopted because it may unintentionally alleviate the harm caused by UV lights for two reasons:

- 1) Encapsulation is often applied to enhance protection. These encapsulation materials (e.g., UV-curable epoxy, polyisobutylene, glass, ethylene vinyl acetate,

- ionomer) often effectively isolate UV lights (either by absorption or reflection);
- 2) The light sources for the long-time MPP testing are usually inorganic light sources that contain much less UV light in the spectrum (*Science* **384**, 1126-1134 (2024)).

Therefore, the MPP testing exhibits a longer device lifetime. For your reference, the ammonium ligand-passivated solar cells here can realize a T_{82} of 1400 h (unpublished data, used as a control device in another work in preparation), confirming its quality.

To summarize, the seemingly rapid performance drop is not only because of illumination but is a result of combined illumination and dynamic electric field stress. The performance drop is attributed to a defect generation process exacerbated by the depassivation mechanism.

We sincerely hope the above reply addresses your concern.

Comment #6. Another intriguing phenomenon: during the light/dark cycling test, the efficiency of the perovskite solar cells increased significantly under both UV-filtered and UV-retained light, but the increase was greater under UV-filtered light. Why is this? Additionally, what was the initial efficiency of the devices used for the light/dark cycling test?

Response: Thank you for your insightful comment.

The efficiency evolution during the cyclic test is a **result of the competition between the light-induced defect annihilation (annihilation of $I_i^-V_i^+$ Frenkel pairs) and light-induced defect generation (two I_i^0 combining into I_2) processes** (*Nat. Photonics* **13**, 532-539 (2019)). The defect generation process tends to occur in a defective region where I is poorly stabilized. In the filtered device, the depassivation of ammonium ligands is not triggered, leaving I sites better stabilized, so the defect annihilation process is favored, improving the efficiency. In contrast, in the unfiltered device, the depassivation mechanism is activated by violet/ultraviolet photons to disrupt the ligand-perovskite interaction. Therefore, more I sites are exposed,

exacerbating the defect generation process. Consequently, the unfiltered device shows a slower efficiency increase initially, as well as a faster efficiency drop later.

This clarification was embedded in Supplementary Text 5. In the revised version, we have added the following text in the main text to clarify (on page 20):

“...The harm caused by violet/UV photons to HP solar cells can occur in several ways. To isolate the effect of the depassivation mechanism here, we subjected the passivated solar cells to a fatigue test with intermittent light-dark alternation. A detailed discussion can be found in Supplementary Text 5. During each replication, the solar cell was first illuminated for an I-V scan for 7 seconds, and then allowed to recover in the dark for 3 seconds for the self-healing of metastable states.^[51] Illumination will evoke a defect-annihilation process ($I_i^- - V_i^+$ Frenkel pairs annihilation) and a defect generation process (two I_i^0 combining into I_2), and the defect generation process tends to occur in a defective region where I^- is poorly stabilized.^[48, 52] The built-in potentials of the heterojunctions vary during the I-V scan in each replication, causing periodic changes in the built-in electric field that affect the ions in the depletion region (fig. S36).”

About the discrepancy in the PCE rise (on pages 20-21):

“...As shown in Fig. 4E, for both illumination conditions, the PCEs increase after the initial several replications, indicating the dominance of the defect annihilation process initially.^[48, 52] The device subjected to the unfiltered illumination manifests a weaker PCE increase, which implies a less favored defect annihilation process. This result is consistent with the violet/UV light-induced depassivation mechanism, *i.e.*, the ligand anchoring for I^- sites is disrupted, so the defect generation process is exacerbated. Indeed, the absorption spectrum characterization throughout this test reveals a significantly more pronounced decline in light absorption for the unfiltered device (fig. S37), which can be ascribed to more pronounced defect generation. Subsequently, the PCE starts to decline as the defect formation process gradually takes over. The decline happens first in the unfiltered device. After 47 replications, the device under the filtered illumination retains 98.0% of its initial PCE, while the value significantly drops to 91.4% for the unfiltered device...”

Figure R14. The I-V characteristics and initial device performances of the solar cell devices subjected to (a) the non-filtered, full-spectrum simulated AM 1.5G solar illumination, and (b) the simulated AM 1.5G solar illumination with photons of <420 nm wavelength filtered.

The initial efficiencies of the solar cell devices subjected to the fatigue test are 23.12% for the non-filtered illumination and 22.31% for the filtered illumination, as shown in Figure R14. They were fabricated from one batch to ensure a fair comparison, despite a slight discrepancy in performance.

We sincerely hope the above reply addresses your concern.

Comment #7. It is mentioned in the text that the depassivation mechanism induced by violet/ultraviolet light is generally applicable to ammonium-based ligands, but currently, near-ultraviolet light bleaching (PB) signals have only been observed in a few 2D perovskites and quasi-2D systems such as BA₂PbI₄. It is suggested to supplement more 3D perovskite data of ammonium ligands with different structures (such as those containing aromatic rings, long-chain alkyl groups, etc.), especially the direct evidence of electron transfer at the ligand-perovskite interface in the pure 3D MAPbI₃ system, to confirm the universality of the mechanism in different dimensions and ligand structures.

Response: Thank you very much for your insightful comment.

We would like to clarify that we have explicitly shown the near-ultraviolet light bleaching signals in a series of $n=1$ 2D perovskites constructed with different organic ligands, including butylammonium, hexylammonium (HA), octylammonium (OA), dodecylammonium (DA), phenylethylammonium (PEA), and phenylpropylammonium (PPA). These ammonium ligands include aliphatic ammoniums and aromatic ammoniums, and they feature different lengths of the aliphatic and aromatic groups. Besides, we have also detected the PB signals in all their mixed- n quasi-2D perovskite systems, except dodecylammonium whose hydrophobicity inhibits the formation of its quasi-2D perovskite film. **The data have all been shown in the Supplementary Information file, Figure S6~S10 for $n=1$ series, Figure S28-S31 for mixed- n systems. Please kindly refer to the Supporting Information file for details.**

The reason we did this research with 2D perovskite systems is that 3D-ligand systems offer surface ligand density, making the detection of this mechanism challenging. When trying to increase the surface ligand density, 3D perovskites easily transition into (quasi-) 2D perovskites to interfere with the interpretation of the results. The 3D-to-2D transition can even be exacerbated in the sample for the transient absorption spectrum, because the MAPbI₃ film sample for the transient absorption spectrum measurement has to be very thin, which facilitates the 3D-to-2D transition. Therefore, preventing the formation of 2D perovskites requires a prudent control over the ligand dose. We have provided this statement in the previous manuscript version (on page 6):

“...To effectively investigate the proposed mechanism, a sample with a high density of well-ordered ammonium-HP bonds is essential. The ammonium-passivated 3D HP system is not an ideal platform due to the sparse and disordered distribution of the ligands on the 3D HP surface.^[32] Instead, we utilized Ruddlesden-Popper 2D HPs with the formula L₂PbI₄, where L represents ammonium-terminated ligands...”

Even for 3D-ligand systems without any 2D perovskite side-products, detecting the photobleaching signal can be challenging. This is because the energy of the ligand-to-perovskite transition overlaps with that of the high-energy absorption of pure 3D MAPbI₃, making the interpretation of the signal difficult. The corresponding statement and data (Figure S34) in the prior manuscript version are shown below (on page 18):

“...Notably, resolving the ammonium-to-HP electron transition signal in surface-treated 3D HP films using TA characterization is challenging due to its overlap with the intrinsic high-energy absorption signals of 3D HPs, as well as difficulties in eliminating interference from 2D HPs (fig. S34)...”

Fig. S34. TA spectra of 3D MAPbI₃ HP films treated with BAI. The surface treatment with ammonium ligands can easily cause the formation of 2D HPs, which complicates the interpretation of the PB signals. The formation of 2D HPs is even exacerbated in the 3D HP film for the TA measurement, as the film needs to be very thin, so the 3D-to-2D conversion is easier to occur. It was not until the concentration of BAI was reduced to 0.3 mg/ml that the formation of 2D HPs was prevented. However, MAPbI₃ itself exhibits high-energy absorption signals, one of which broadly distributes in the ultraviolet region and overlaps with the ammonium-HP interaction signal. This overlap makes it challenging to resolve the ammonium-HP interaction signal in ammonium-passivated 3D HPs using TA spectrum measurements.

Although we are unable to provide the transient absorption spectrum characterization in 3D perovskite systems due to the abovementioned challenges, we have provided steady-state UV-vis absorption spectra to verify the mechanism. Take the BA surface-

treated MAPbI₃ film as an example, we observe an absorption enhancement in the 300 nm - 400 nm range, peaked at around 350 nm (3.10 eV - 4.13 eV, peaked around 3.54 eV). This absorption enhances with increasing BA concentration. The energy does not match the intrinsic absorption of BA itself (Fig. S3 in the manuscript) but agrees with the depassivation mechanism:

Figure R15. (Figure S32a in the manuscript). (a) Steady-state absorption spectra of 3D HP (MAPbI₃) films treated by BAI solutions with various concentrations on the surface.

As shown in Figure R16 below, this phenomenon is also observed in many other ammonium ligands, including hexylammonium, octylammonium, dodecylammonium, phenylethylammonium, and phenylpropylammonium. These ligands include aliphatic and aromatic types with a series of alkyl/aromatic chain lengths, verifying universality.

Figure R16 (Figure S33 in the manuscript). The steady-state absorption spectra of 3D HP MAPbI₃ before and after surface treatments with various ammonium ligands, including (a) PPA (in the form of PPAI), (b) PEA (in the form of PEAI), (c) DA (in the form of DAI), (d) OA (in the form of OAI), (e) and HA (in the form of HAI). Insets: Zoom-in regions of absorption spectra in the range of 300-400 nm.

We can assess the validity of the energy range of the enhanced absorption by drawing a comparison with BA₂PbI₄. In BA₂PbI₄, the butylammonium (BA⁺) ligands are an integral part of the crystal structure, orderly arranged and strongly interacting with the inorganic [PbI₆]⁴⁻ perovskite layers through electrostatic forces and hydrogen bonding. This strong, highly ordered environment leads to an effective HOMO of the BA⁺ within this 2D system reported at -6.5 eV relative to the vacuum level. In contrast, the attachment of BA⁺ ligands to the surface of 3D MAPbI₃ perovskites is typically more random and dynamic. This implies a less extensive and less structured interaction between the -NH₃⁺ group of the ligand and the underlying 3D perovskite surface, compared to its role as a structural spacer in 2D perovskites. This reduced perturbation from the perovskite lattice allows the N-H bonds within the BA⁺ cation to retain more of their intrinsic strength. Consequently, the electrons in the HOMO of the ammonium group are held more tightly within the ligand itself. **This leads to a deeper (more negative) HOMO energy level for the ammonium group when adsorbed on the 3D MAPbI₃ surface**, compared to its more delocalized and stabilized state within the highly ordered 2D BA₂PbI₄ lattice. Considering that the typical CBM of MAPbI₃ films is -3.9 eV versus the vacuum level, and the absorption peak revealed by the absorption

spectra is 3.5 eV, we can estimate the HOMO level of the ammonium group on the 3D perovskite surface to be -7.4 eV. **This value is indeed more negative than that in BA₂PbI₄ (-6.5 eV)**, which aligns with our reasoning. Therefore, ascribing the enhanced absorption behavior to the ammonium-to-perovskite carrier transition is tenable.

To conclude, directly observing the ammonium-to-perovskite carrier transition in the ligand-3D perovskite system with the transient absorption spectrum characterization is challenging. This is because the dose of the ammonium ligands needs to be very low to prevent the interference of 2D perovskites, which easily form during the surface treatment. The low ligand coverage on the 3D perovskite surface weakens the photobleaching signal. Besides, the photobleaching signal overlaps with the intrinsic high-energy absorption of the 3D perovskite film to obscure the data interpretation. However, confirmed in n=1 and mixed-n 2D perovskite systems, the mechanism is in principle extendible to 3D perovskites. Experimentally, the steady-state absorption spectrum characterization shows an enhanced absorption in the ultraviolet region after the 3D perovskite film is surface-treated with ammonium ligands. The absorption enhancement strengthens as the ligand concentration increases. This is observed for various ammonium ligands, confirming universality. The enhanced absorption and the energy range align with the mechanism here. In the manuscript, follow-up characterizations, including FTIR and device characterizations, all agree with the mechanism reported here.

We sincerely hope the above reply addresses your concern. Thank you very much for your understanding.

We sincerely hope the above reply addresses your concern.

Comment #8. The near-ultraviolet PB signal is attributed to the electron transfer from ammonium to perovskite, but the influence of impurities (such as unreacted PbI₂ or ligand decomposition products) is not completely excluded in the manuscript. Although XRD shows that the purity of some samples is relatively high, it is necessary to further confirm through characterization methods such as high-resolution transmission electron microscopy (HRTEM) or time-of-flight

secondary ion mass spectrometry (TOF-SIMS) that the signals only come from the ligand-perovskite interface, rather than residual precursors or degradation products.

Response: Thank you for your comment.

We agree with the necessity to strengthen the conclusion that the near-UV photobleaching signal is from 2D perovskites rather than potential impurities. We agree that HRSTEM and TOF-SIMS can help understand the major components of the samples, but respectfully, these characterizations may be less convincing in excluding impurities. For example, even if an HRSTEM image is present to show high phase purity, the conclusion only applies to this specific region of observation, and it is impractical to exclude all potential impurities in the entire film. This is why we adopted the XRD characterization, as it provides a comprehensive analysis of the entire film. Furthermore, in response to your comment, we have added the following analysis to further strengthen the argument.

Figure R17. (a) Pseudocolor ultrafast TA spectrum of the BA₂PbI₄ film under 3.82 eV (325 nm, 1 kHz, 100 fs, ~510 $\mu\text{W}/\text{cm}^2$) laser excitation, showing two near-UV PB signals in addition to the excitonic PB signal. (b) Schematic illustration of the electronic structure of BA₂PbI₄ with a highlight of key energy levels relative to the vacuum level.

The transient absorption spectrum characterization can provide additional details to confirm that the near-UV photobleaching signals are truly from 2D perovskites themselves. In the previous version of transient absorption spectra, we only showed the photobleaching signals caused by the HOMO-to-E_x (E_x refers to the excitonic state) carrier transition. In fact, two photobleaching signals can be identified, and the other

one matches the HOMO-to-CBM transition perfectly.

Take BA_2PbI_4 for an example, the detailed TA spectrum is shown in Figure R17a. In addition to the excitonic absorption photobleaching in the range of 2.43-2.57 eV, there are **two** photobleaching signals in the near-UV region. The low-energy one at 2.90-3.10 eV was introduced in the prior manuscript version. It corresponds to the HOMO-to- E_x carrier transition. In addition, there is another high-energy PB signal centered at ~ 3.3 eV. The energy level distribution of the three photobleaching states well matches the energy band structure of BA_2PbI_4 , as shown in Figure R17b.

The two PB signals in the near-UV region are also observed in a series of other 2D perovskites constructed with various ammonium-terminated ligands (hexylammonium (HA), octylammonium (OA), dodecylammonium (DA), phenylethylammonium (PEA), and phenylpropylammonium (PPA)), as shown in Figure R18 below:

Figure R18. Pseudocolor ultrafast TA spectrum of a series of 2D perovskites constructed with various ammonium-terminated ligands. In all these samples, two PB signals in the near-UV region can be identified. These 2D perovskites are (a) PPA_2PbI_4 , (b) PEA_2PbI_4 , (c) DA_2PbI_4 , (d) OA_2PbI_4 , (e) HA_2PbI_4 . Notably, the signals in DA_2PbI_4 are relatively weak, which can be ascribed to a less crystalline nature of the film due to the ultrahigh hydrophobicity of DA.

The observation of two PB signals with their energies closely matching the HOMO-to- E_x carrier transition and the HOMO-to-CBM carrier transition, along with the XRD

characterization, provides strong support that the PB signals are from 2D perovskites themselves. Accordingly, we have revised the corresponding statement in the manuscript (on page 8), as shown below:

“Based on the energy diagram, the energies of the two near-PB signals (2.9 - 3.1 eV for the low-energy one and 3.2 - 3.4 eV for the higher-energy one) closely match those of the HOMO-to- E_x (3.1 eV) and HOMO-to-CBM (3.4 eV) carrier transitions, respectively. The minor deviations are reasonable considering that the DFT calculation adopted for the HOMO in the previous report usually overvalues the result. A supportive piece of evidence might be the emergence of the onset of a new band in the Fermi region of the UPS spectrum at approximately 0.43 eV below the VBM (fig. S11), which can tentatively be attributed to the HOMO. As such, the true HOMO might be within the range of -6.25 eV (-5.82 eV minus 0.43 eV) to -6.50 eV versus the vacuum level, so the HOMO- E_x gap and the HOMO-CBM gap could be in the range of 2.85 - 3.10 eV and 3.15 - 3.40 eV, justifying the above deviations. Notably, the HOMO-to- E_x and HOMO-to-CBM transitions are also consistent with the above observation of anti-Stokes PB signals.”

We sincerely hope the above reply addresses your concern. Thank you for your kind understanding.

We sincerely hope the above reply addresses your concern.

Comment #9. In the manuscript, the HOMO/LUMO energy levels of ligands are defined based on the DFT calculation results of Silver et al. However, different calculation methods (such as functional selection and basis set size) may lead to deviations in energy values. It is suggested to supplement the first-principles calculations for specific ligands (such as BA^+), clarify the consistency degree between the energy difference of HOMO-exciton (E_x) and the energy of the near-ultraviolet PB signal observed experimentally, and discuss the possible sources of the deviation between the calculation and the experiment (such as solvent effect,

exciton binding energy).

Response: Thank you very much for your comment.

There is very limited report on the detailed electronic structure of BA₂PbI₄ 2D perovskites, and the reference (Silver, S. et al. *Adv. Energy Mater.* 8, 1703468 (2018)) we provided in the manuscript is the only report we found that offers details on it, especially the HOMO level position. This work is widely cited in the 2D perovskite research community (115 times cited), verifying wide acknowledgement. The limited source we can access does not allow us to provide a more precise simulation than this work.

However, we can provide some additional experimental evidence and analysis to support the claim that the observed near-UV PB signal corresponds to the HOMO-to-Ex carrier transition, as listed below:

- 1) First, we detailed in the response to the above comment (comment #8). There are actually two near-UV PB signals. For the archetypal BA₂PbI₄ sample exhibited in the main text, the lower-energy one is in the 2.90 - 3.10 eV range, and the higher-energy one is in the 3.20 - 3.40 eV range. The energy gap between them matches the reported exciton binding energy range for BA₂PbI₄ (e.g., **260 meV** reported in Silver, S. et al. *Adv. Energy Mater.* 8, 1703468 (2018); **380 meV** reported in Hong, H. L. et al. *Nat. Commun.* 15 (2024)).
- 2) According to the electronic structure of BA₂PbI₄, the two near-UV PB signals closely match the HOMO-to-E_x and HOMO-to-CBM transitions, respectively. The electronic structure suggests a HOMO-to-E_x gap of 3.1 eV and a HOMO-to-CBM gap of 3.4 eV. The two values reach the upper end of the energies of the two PB signals observed. The deviations can be ascribed to an overvaluation of the HOMO position common for DFT simulations (deeper than the actual value), and might be justified by our UPS characterization. As shown in Figure R19 below, the UPS spectrum reveals the emergence of the onset of a new band at approximately 0.43 eV below the VBM, close to but less deep than the simulated HOMO level. As such, the HOMO-E_x gap and the HOMO-CBM gap could be in the range of 2.85 - 3.10 eV and 3.15 - 3.40 eV, justifying the above deviations.

Figure 19 (Figure S11 in the revised manuscript). UPS characterization of the BA_2PbI_4 film. (a) Fermi region and (b) secondary electron region of the UPS spectrum of BA_2PbI_4 . (c) Identification of the possible VBM-HOMO energy gap in the logarithmic scale of the Fermi region of the UPS spectrum.

Based on the above analysis, we have strengthened the corresponding argument in the main text (on page 8), as shown below:

“...To better understand the origin of the near-UV PB signal, we quantitatively depict the electronic structure of BA_2PbI_4 with a highlight of key energy levels, including the CBM, E_x , VBM, lowest unoccupied molecular orbital (LUMO) of the ligand layer, and highest occupied molecular orbital (HOMO) of the ligand layer, in Fig. 1D. The VBM is measured with Ultraviolet Photoelectron Spectroscopy (UPS) characterization (fig. S11). The obtained VBM (-5.82 eV) and the accordingly inferred CBM based on a bandgap of 2.7 eV agree with the previous report by Silver *et al.*^[31] E_x is determined per the excitonic absorption energy (fig. S2, 2.41 eV). The HOMO and LUMO also refer to the report by Silver *et al.* obtained with density functional theory (DFT) calculations.^[31] Based on the energy diagram, the energies of the two near-PB signals (2.9 - 3.1 eV for the low-energy one and 3.2 - 3.4 eV for the higher-energy one) closely match those of the HOMO-to- E_x (3.1 eV) and HOMO-to-CBM (3.4 eV) carrier transitions, respectively. The minor deviations are reasonable considering that the DFT calculation adopted for the HOMO in the previous report usually overvalues the result. A supportive piece of evidence might be the emergence of the onset of a new band in the Fermi region of the UPS spectrum at approximately 0.43 eV below the VBM (fig. S11), which can tentatively be attributed to the HOMO. As such, the true HOMO might be within the range of -6.25 eV (-5.82 eV minus 0.43 eV) to -6.50 eV versus the vacuum level, so the HOMO- E_x gap and the HOMO-CBM gap could be in the range of 2.85 - 3.10 eV and 3.15 - 3.40 eV, justifying the above deviations...”

References

31. Silver, S. *et al.* Characterization of the valence and conduction band Levels of $n = 1$ 2D perovskites: a combined experimental and theoretical investigation. *Adv. Energy Mater.* **8**, 1703468 (2018).

We sincerely hope the above reply addresses your concern.

Comment #10. Figure 4E shows that the device efficiency decreases under ultraviolet light, but the current data only infer the formation of defects through the I-V curve. It is suggested to supplement in-situ characterization methods, such as time-resolved photoluminescence (TRPL) under light or spatially resolved surface photovoltage spectroscopy (SPS), to directly monitor the density changes of non-radiative recombination centers caused by ligand depassivation, and establish the real-time correlation between photocurrent attenuation and ammonium dissociation.

Response: Thank you for your insightful comment. We agree that the suggested evidence will strengthen the argument in our manuscript.

We have carried out the TRPL test as suggested. However, the rapid degradation of the device during the TRPL test significantly interferes with the result interpretation. The details of the test are as shown below: In order not to activate the depassivation mechanism during the TRPL test, we used a 450 nm excitation wavelength. The excitation power is $\sim 70 \mu\text{W}$. The device was in a configuration of ITO/4PADCB/ Al_2O_3 / $\text{Cs}_{0.05}\text{FA}_{0.85}\text{MA}_{0.1}\text{PbI}_3$ /LiF/ C_{60} /BCP/Ag, the same device configuration for the aging test. For the TRPL test, the device was tested unencapsulated, as encapsulation materials usually screen violet/ultraviolet photons, which is undesired in our control experiment. In our measurement, **the device performance dramatically declines after only one round of the TRPL test.** The TRPL result and the change in the I-V characteristic before and after the TRPL test are shown in Figure R20 below:

Figure R20. (a) TRPL test result of the device. (b) I-V characteristics of the device measured before and after the TRPL test.

The severe performance degradation is expected to arise from the synergetic effect imposed by the excitation light and the ambient air. Based on this observation, we are afraid this degradation would interfere with the analysis of the depassivation mechanism in this work.

Alternatively, we measured the **absorption spectra** of the solar cell device during the light-dark alteration test. Compared with the TRPL test, the absorption spectrum measurement is milder and more friendly for the devices, preventing damage as in the TRPL test.

For each cycle during the ageing test, the device was subjected to a reverse scan from 1.2 V to -0.05 V under illumination, and was then followed by a recovery in the dark. The illumination time is 7 s, and the dark recovery time is 3 s. Two illumination conditions were applied for comparison: one is non-filtered, full-spectrum simulated AM 1.5G solar illumination, and the other is filtered simulated AM 1.5G solar illumination that eliminates photons of wavelength below 420 nm.

Figure R21. Evolutions of the baseline-corrected absorption spectra of solar cell devices subjected to (a) the non-filtered, full-spectrum illumination and (b) the filtered illumination, respectively, during the fatigue test.

We measured the evolution of the absorption spectrum of solar cell devices subjected to the two testing conditions. Due to the high reflectivity enabled by the Ag electrode, the overall baseline of the absorption is high. But still, the absorption feature of the perovskite layer can be identified for analysis. The baseline-corrected absorption spectra are shown in Figure R21. For the device subjected to the non-filtered, full-spectrum simulated AM 1.5G solar irradiation, which contains photons with wavelengths below 420 nm that can trigger the depassivation mechanism, a severe degradation of the absorption feature is observed as the fatigue test proceeds. This can be ascribed to the ion migration driven by the varying built-in electric field during the fatigue test, which exacerbates defect generation to compromise the light absorption. In stark contrast, the absorption of the device subjected to the filtered illumination shows a significantly mitigated degradation feature. This result is in line with the depassivation mechanism, as one of the key effects of ammonium ligands is to anchor halide ions.

Accordingly, we have added this discussion in the revised manuscript (on pages 20-21) to strengthen the argument, as follows:

“We used two illumination conditions for comparison: violet/UV light-filtered solar illumination (using a >420 nm long-pass filter) and unfiltered illumination. The solar

simulator we used was measured to contain 12.78% of its total irradiance at wavelengths below 420 nm, slightly higher than that in the standard AM 1.5G irradiation (~5%). The filtered light intensity was calibrated to compensate for the filtering loss. Besides, to exclude interference from ambient factors, we conducted the tests in an N₂-filled glovebox. As shown in Fig. 4E, for both illumination conditions, the PCEs increase after the initial several replications, indicating the dominance of the defect annihilation process initially.^[48, 52] The device subjected to the unfiltered illumination manifests a weaker PCE increase, which implies a less favored defect annihilation process. This result is consistent with the violet/UV light-induced depassivation mechanism, *i.e.*, the ligand anchoring for Γ sites is disrupted, so the defect generation process is exacerbated. Indeed, the absorption spectrum characterization throughout this test reveals a significantly more pronounced decline in light absorption for the unfiltered device (fig. S37), which can be ascribed to more pronounced defect generation. Subsequently, the PCE starts to decline as the defect formation process gradually takes over. The decline happens first in the unfiltered device. After 47 replications, the device under the filtered illumination retains 98.0% of its initial PCE, while the value significantly drops to 91.4% for the unfiltered device.”

References

- 48 Motti, S. G. *et al.* Controlling competing photochemical reactions stabilizes perovskite solar cells. *Nat. Photonics* **13**, 532-539 (2019).
- 52 Mosconi, E. *et al.* Light-induced annihilation of Frenkel defects in organo-lead halide perovskites. *Energy Environ. Sci.* **9**, 3180-3187 (2016).

We sincerely hope the above reply addresses your concern.

Comment #11. In the light reaction experiment, the increase in the generation rate of H₂ indicates the deprotonation of the ammonium group, but there is a lack of kinetic curves under different light intensities (such as the variation of H₂ generation amount with time). It is suggested to supplement the ultraviolet light intensity gradient experiment to quantify the relationship between photon flux and deprotonation rate, and analyze the activation energy of this process through the Arrhenius equation to distinguish light-induced electron transfer from

thermally driven ligand decomposition.

Response: Thank you for your insightful comment. We totally agree that the thermally driven ligand decomposition should be excluded to strengthen the argument. In fact, we would like to clarify that the H₂ generation experiment was implemented at a fixed temperature (10 degrees Celsius) throughout the measurement. We feel sorry for not including this detail in the experimental section. Taking measurements at a fixed temperature is a routine practice in the photocatalysis-related experiments, and its aim is exactly to exclude the impact of heating.

Accordingly, we have added this detail to the revised manuscript (on page 15), as shown below:

“Accordingly, the production of H₂ is a direct result of the dynamic deprotonation of the ammonium groups. The vessel used for the photoreactions was **evacuated** to minimize interference from other potential oxidants and reductants. **The temperature of the vessel was fixed at 10 degrees Celsius during the test to prevent interference from heating.** The 2D HP is stable throughout the test as discussed in the Supporting Information file (fig. S24 and Supplementary Text 2).”

The precise measurement of the photon flux and the deprotonation rate, while surely would contribute to the discussion, can be challenging in practice. In this experiment, 2D perovskites in the form of nanoplates disperse in an HI aqueous solution, which is in a glass container (vacuumed and temperature fixed). Illumination is from an external light source, so incident photons have to pass the glass container (reflection), be absorbed by the HI aqueous solution (absorbs photons with a wavelength below 400 nm), and be scattered by the 2D perovskite nanoplates themselves before being absorbed by the 2D perovskites. These processes are not linear with intensity.

Figure R22. EQEs of the photocatalytic hydrogen production upon the 365 nm irradiation and the 500 nm irradiation, respectively.

Alternatively, to strengthen the argument, we measured the external quantum efficiency (EQE) of the photocatalytic H₂ generation upon monochromatic irradiations of 365 nm wavelength and 500 nm wavelength, respectively. The 365 nm is energetically sufficient to trigger the deprotonation mechanism, while the 500 nm irradiation resonates with the absorption of the excitonic absorption of 2D perovskites. EQE is the ratio of the number of useful reaction events (H₂ generation in this case) by a photocatalyst to the total number of photons that strike it from an incident light source. Generally, UV photons above the bandgap should deliver a lower EQE due to thermal energy loss, and UV photons are more prone to loss by the abovementioned adverse processes (reflection, absorption, scattering). However, we found that the 365 nm irradiation still delivers a much higher EQE (43.4%) than the 500 nm irradiation (23.2%), which aligns with the photon-induced deprotonation effect in our work. The residual H₂ generation by the 500 nm excitation arises from the direct exciton transfer from 2D perovskites to H⁺/I⁻ ions, as detailed in the manuscript.

We sincerely hope the above reply addresses your concern.

Comment #12. Fatigue tests were conducted in a nitrogen glove box, but in practical applications, water vapor and oxygen may accelerate the degradation of perovskite. It is suggested to supplement the comparative experiments in the air environment to clarify whether the depassivation induced by violet/ultraviolet light is independent of environmental factors or has a synergistic effect with water vapor. Furthermore, it is necessary to explain whether the intensity of the ultraviolet light used in the test matches the ultraviolet component in the actual sunlight (for example, the ultraviolet proportion in the AM 1.5G spectrum is approximately 5%).

Response: Thank you for your comment.

The light-induced depassivation mechanism should have a synergetic effect with the light-assisted erosion of oxygen/moisture in the air. This is because the violet-ultraviolet photons will pump the ammonium-to-perovskite electron transition, potentially exacerbating the redox erosion by oxygen/moisture in the air to boost the depassivation process.

Experimentally, we used solar cells with the configuration ITO/4PADCB/Al₂O₃/Cs_{0.05}FA_{0.85}MA_{0.1}PbI₃/LiF/C₆₀/BCP/Ag. It should be noted that this is the same proof-of-concept device configuration in the manuscript, but in our previous version, the device configuration is mistakenly written as ITO/NiO_x/Meo-4PACZ/Cs_{0.05}FA_{0.85}MA_{0.1}PbI₃/LiF/C₆₀/BCP/Ag due to multi-person editing, and we are so sorry for this careless mistake, but it does not impact the core of this work. The perovskite layer is passivated with octane-1,8-diammonium and phenylpropylammonium. One cell was subjected to 365 nm irradiation in the air for 30 min. The irradiation power was 6.4 mW/cm². Another solar cell was subjected to 450 nm irradiation in the air with the same irradiation period and irradiation intensity. The 365 nm irradiation can activate the depassivation mechanism, while the 450 nm irradiation cannot. After the irradiation test, both samples were stored in the glovebox in the dark for 5 h's recovery. The recovery process eliminates the impact of metastable trap states.

Figure R23. (a) Initial and after-ageing I-V characteristics of the solar cell subjected to the 365 nm irradiation test. (b) Initial and after-ageing I-V characteristics of the solar cell subjected to the 450 nm irradiation test. (c) The EQE spectra of the device in (a) before and after ageing. (d) The EQE spectra of the device in (b) before and after ageing.

Table R1. Performance metrics of solar cells before and after ageing

Irradiation	Sample	J_{sc} (mA/cm ²)	V_{oc} (V)	FF (%)	PCE (%)
365 nm	Initial	22.70	1.140	85.16	22.04
	Ageing	22.80	1.091	69.98	17.41
450 nm	Initial	23.31	1.143	85.19	22.70
	Aging	23.41	1.097	76.60	19.67

The device performance before and after the ageing process is shown in Figure R23 and Table R1. Both ageing processes cause an efficiency drop. Notably, the 365 nm irradiation causes a significant decline of FF from 85.16% to 69.98%. In contrast, the device subjected to the 450 nm irradiation only manifests an FF drop from 85.19% to

76.60%. This result indicates the emergence of additional shunts in the 365 nm-irradiated device, which is in line with the depassivation mechanism proposed. The J_{sc} value does not change conspicuously for both devices before and after ageing, and the V_{oc} value drops are almost identical.

For the 365 nm-irradiated device, the EQE spectra show that the EQE values in the short-wavelength region decrease and those in the long-wavelength region increase after the ageing. This could arise from phase segregation that generates wider-bandgap phases near the bottom and narrower-bandgap phases in the bulk. In contrast, for the 450 nm-irradiated device, the EQE profile barely changes before and after ageing. This contrast is consistent with the depassivation mechanism. That is, the ammonium-ligands fail to anchor ions upon the 365 nm irradiation to facilitate the phase segregation.

Based on the above analysis, we infer that the light-induced depassivation mechanism should have a synergetic effect with the erosion by oxygen and moisture in the air.

The relative contribution of photons with wavelengths shorter than 420 nm to the total irradiance of the solar simulator in our lab, a Newport 94023A solar simulator, was determined using a Newport 91150V reference cell and a Thorlabs FGL420M long-pass filter. Based on our measurement, the Newport 94023A solar simulator produces approximately **12.78% of its total irradiance at wavelengths below 420 nm**, equivalent to 7.95% Suns under the photoresponse of the Newport 91150V reference cell. In standard AM 1.5 G solar irradiation, the irradiance should indeed be ~5%, as you commented. Our solar simulator for the solar cell device test contains a higher proportion of the irradiance at wavelengths below 420 nm. The effective inclusion of this violet/ultraviolet irradiation makes the comparative solar cell characterization convincing. Although this might cause deviation in measuring the device performance, the contribution from the violet/ultraviolet part is relatively minor. Besides, we have also provided certificated solar cell performance (Fig. S36 in the Supplementary Information file) to validate the device quality for this study.

To clarify the irradiance proportion in the solar simulator we used, we have added the following clarification in the revised manuscript (on page 20):

“We used two illumination conditions for comparison: violet/UV light-filtered solar illumination (using a >420 nm long-pass filter) and unfiltered illumination. **The solar**

simulator we used was measured to contain 12.78% of its total irradiance at wavelengths below 420 nm, slightly higher than that in the standard AM 1.5G irradiation (~5%).”

We sincerely hope the above reply addresses your concern.

Comment #13. The FTIR results show that the ligand signal weakens after ultraviolet irradiation, but it is only a static characterization. It is recommended to use in situ FTIR to monitor the real-time changes of ammonium characteristic peaks (such as N-H scaling vibration) during the irradiation process, clarify the dynamic path and reversibility of deprotonation ($-\text{NH}_3^+ \rightarrow -\text{NH}_2$), and rule out the possibility of physical desorption of ligands.

Response: Thank you for your insightful comment. Following your comment, we have measured the irradiation time-dependent FTIR spectra. The details are shown below:

We selected MAPbI₃ as the perovskite sample, as it is single-anion, single-cation, and phase-stable at room temperature. Otherwise, there could be a phase segregation or a phase transition that interferes with the interpretation. The ligand pair butylammonium and oleic acid was used. 100 mg MAPbI₃ polycrystalline powder was dispersed in a 5 mL chlorobenzene solution containing 11.9 mg butylamine and 45.8 mg oleic acid (1:1 in molar ratio). The solution was gently stirred with a magnetic stirrer during the test. In the meantime, we applied a monochromatic illumination of 365 nm to the solution with a power density of ~25 mW/cm². The 365 nm irradiation can trigger the depassivation mechanism. We extracted the dispersion every hour and isolated the MAPbI₃ powders by centrifugation and vacuum drying. In comparison, the same MAPbI₃ dispersion was subjected to the same test except that it was covered with aluminum foil to avoid irradiation. The irradiated sample with different irradiation times, the non-irradiated sample, and the pristine MAPbI₃ sample were sent for FTIR measurement.

Figure R24. The irradiation time-dependent FTIR spectra of the passivated perovskite sample.

The FTIR characterization confirms the gradual detachment of ligands under the 365 nm irradiation. As shown in Figure R24, the 2854 cm^{-1} peak arises from the symmetric stretching vibration of $-\text{CH}_2-$ groups and can originate from both the butylammonium and oleic acid segments in the ligand pair, while it is not detected in non-passivated, pristine MAPbI_3 . Notably, the characteristic peak of the N-H scaling vibration may not be an ideal observation target, as non-passivated MAPbI_3 perovskite intrinsically shows a strong N-H scaling vibration signal as well. Importantly, as the irradiation time increases, this characterization peak wanes gradually, indicating the detachment of the ligands from the perovskite surface. This is in line with the photoinduced depassivation mechanism introduced in our work. After 5 h of irradiation, this characteristic peak becomes barely discernible. In contrast, for the non-irradiated sample, the 2854 cm^{-1} peak remains after the 5 h test. This contrast rules out the impact of physical detachment and confirms that the 365 nm irradiation is the driving factor of the ligand detachment, agreeing with the photoinduced detachment mechanism.

Accordingly, we have made modifications in the revised manuscript (on pages 18-19),

as shown below:

“Consequently, exposure to violet/UV photons causes the deprotonation of ammonium ligands, disrupting the ammonium-HP interaction. To verify this, we conducted Fourier-transform infrared (FTIR) spectroscopy on both UV-irradiated (365 nm @ 25 mW/cm² for 5 h) and non-irradiated ammonium-passivated MAPbI₃ samples. During irradiation, the 3D HP powder was gently stirred and dispersed in chlorobenzene with ligands. It is well established that HPs adopt an X-type ligand binding configuration, where Lewis acid and Lewis base moieties attach to the surface of HP in pairs.^[32, 46] Herein, we employed BA and oleic acid (1:1 in molar ratio) as the ligand pair. The 3D HP powder was collected every one hour, followed by isolation for the FTIR characterization. As shown in Fig. 4C, the 2854 cm⁻¹ peak arises from the symmetric stretching vibration of -CH₂- groups. It originates from both the butylammonium and oleic acid segments in the ligand pair but is not detected in MAPbI₃. Notably, as the irradiation time increases, the -CH₂- peak wanes gradually, indicating the detachment of ligands from the HP surface. After 5 h of irradiation, this characteristic peak becomes barely discernible. In contrast, for the non-irradiated sample, the 2854 cm⁻¹ peak remains after the 5h test. This contrast confirms that the 365 nm irradiation is the driving factor for ligand detachment, in agreement with our mechanism.”

References

32. De Roo, J. et al. Highly dynamic ligand binding and light absorption coefficient of cesium lead bromide perovskite nanocrystals. *ACS Nano* **10**, 2071-81 (2016).
46. Nenon, D. P. et al. Design principles for trap-free CsPbX₃ nanocrystals: enumerating and eliminating surface halide vacancies with softer lewis bases. *J. Am. Chem. Soc.* **140**, 17760-17772 (2018).

In terms of the dynamic deprotonation path, we have to admit that we are unable to elucidate it due to its complex nature. Even without considering the interference of air, the ligand dynamics could involve not only the redox reaction of ligand pairs themselves but also ions in the perovskite lattice (halide ions), as follows:

where BA denotes butylammonium, OLA refers to oleic acid. We believe that detailed dynamic paths require detailed exploration, which is unlikely to be incorporated into this work. We sincerely hope for your kind understanding.

In the end, we sincerely thank you again for dedicating your valuable time to reviewing our work and for providing insightful feedback. We hope our responses adequately address your concerns and align our manuscript with the high standards of *Nature Communications*. We have made every effort to refine our work and welcome any further comments or suggestions in subsequent reviews. Thank you once again for your thoughtful contributions.